# Primary care quality for older adults: Practice-based quality measures derived from a RAND/UCLA appropriateness method study

Rebecca H. Correia[1]*, Darly Dash[1], Aaron Jones[1], Meredith Vanstone[2], Komal Aryal[1], Henry Yu-Hin Siu[2], Aquila Gopaul[3], Andrew P. Costa[1]

1 Department of Health Research Methods, Evidence and Impact, Faculty of Health Sciences, McMaster University, Hamilton, Ontario, Canada, 2 Department of Family Medicine, Faculty of Health Sciences, McMaster University, Hamilton, Ontario, Canada, 3 Department of Family Medicine, Western University, London, Ontario, Canada

* correirh@mcmaster.ca

**Data Availability Statement:** All relevant data are within the manuscript and its Supporting information files.

## Abstract

We established consensus on practice-based metrics that characterize quality of care for older primary care patients and can be examined using secondary health administrative data. We conducted a two-round RAND/UCLA Appropriateness Method (RAM) study and recruited 10 Canadian clinicians and researchers with expertise relevant to the primary care of elderly patients. Informed by a literature review, the first RAM round evaluated the appropriateness and importance of candidate quality measures in an online questionnaire. Technical definitions were developed for each endorsed indicator to specify how the indicator could be operationalized using health administrative data. In a virtual synchronous meeting, the expert panel offered feedback on the technical specifications for the endorsed indicators. Panelists then completed a second (final) questionnaire to rate each indicator and corresponding technical definition on the same criteria (appropriateness and importance). We used statistical integration to combine technical expert panelists' judgements and content analysis of open-ended survey responses. Our literature search and internal screening resulted in 61 practice-based quality indicators for rating. We developed technical definitions for indicators endorsed in the first questionnaire (n = 55). Following the virtual synchronous meeting and second questionnaire, we achieved consensus on 12 practice-based quality measures across four Priority Topics in Care of the Elderly. The endorsed indicators provide a framework to characterize practice- and population-level encounters of family physicians delivering care to older patients and will offer insights into the outcomes of their care provision. This study presented a case of soliciting expert feedback to develop measurable practice-based quality indicators that can be examined using administrative data to understand quality of care within population-based data holdings. Future work will refine and operationalize the technical definitions established through this process to examine primary care provision for older adults in a particular context (Ontario, Canada).

**Funding:** RHC is supported by a Canadian Institutes of Health Research Canada Graduate Scholarship (funding reference #181540). MV is supported by a Canada Research Chair (Tier 2) in Ethical Complexity in Primary Care. APC is supported by a Canada Research Chair (Tier 2) in Integrated Care for Seniors. The funders had no role in study design, data collection and analysis, decision to publish, or preparation of the manuscript.

**Competing interests:** The authors have declared that no competing interests exist.

## Introduction

Community-based primary healthcare is often patients' first point-of-contact with health services [1]. Older adults (aged 65+) are the highest users of primary care services [2], and their number of interactions and dependence on health systems increases with age [3]. Aging demographics [4], compounded by older adults' growing social vulnerabilities [5–10] and preference to remain in the community [11, 12], portends even greater demands on primary care systems in the coming years [3, 13]. Primary care is currently grappling with substantial healthcare provider workforce shortfalls [14–16], posing significant challenges for patient access [17, 18]. Improvements in primary care structures and processes are essential to create the necessary capacity for family physicians to adequately address the complex and emerging needs of older adults [1, 19, 20].

In Canada, family physicians are central to primary care teams and are the most frequent physician providers of medical services to older adults [2]. Although all family physicians must achieve a baseline level of competence to care for elderly patients to be certified for independent practice, there is a range of competence across family physicians [21]. Variability in knowledge and clinical skills of providers may influence quality of care and health outcomes for older patients. Therefore, developing indicators to assess, measure, and compare the medical practice structures and services provided by family physicians–in the context of caring for older patients–may inform quality improvement activities and priorities for continuing professional education [22, 23].

Inferences about quality of care are limited by the availability of data since reliable and valid data holdings are needed to apply indicators [23]. Information systems, such as population-based health administrative data sources, enable the accurate and consistent measurement of indicators by providing access to data at the macro (e.g., healthcare organizations) and micro (e.g., individual patients, providers) levels. Research focused on the development of elderly-focused quality indicators for primary care to-date have not been concerned with operationalizing them [21]. In this context, "operationalize" refers to developing technical specifications to apply, measure, and examine indicators using available data holdings. Despite the limitations of using secondary data to make inferences about quality [21], indicators developed without reference to information sources or those that are not feasible to operationalize within a particular context are limited in their utility and capacity to promote change.

Therefore, we sought to establish consensus on measurable practice-based process metrics that characterize quality of care for older primary care patients. We focus on quality indicators that are directly relevant to the clinical activities of physicians (practice-based) and classified as acts of healthcare service delivery (processes). Along with clinical structures and medical outcomes, healthcare service delivery processes are one of three interrelated components comprising the Donabedian model–the dominant quality improvement paradigm that enables evaluations of medical care quality and the performance of health systems [21, 24]. Our study is situated within the province of Ontario, Canada to develop technical definitions for each indicator using available administrative data holdings. We organized our work around the research question: *Within the framework of secondary, administrative data as a lens to understand primary care practice, can a technical expert panel establish consensus on which practice-based process metrics suggest better versus worse quality of care for older patients*? We hypothesized that derived indicators would be clinically meaningful and feasible to measure, but not comprehensive of all clinical activities relevant to older adult-focused primary care. Our endorsed indicator set will support future work examining the medical practice of family physicians and quality of care for older patients.

## Methods

### Study design

We conducted a two-phase RAND/UCLA Appropriateness Method (RAM) study that has been described previously [21]. The Delphi technique and its derivatives, including RAM, have been widely applied in health services research for quality indicator development. We report our study using the CREDES guideline [25] (S1 Table). We describe the four major stages of our study.

### Stage 1: Literature review to identify candidate indicators

We conducted a literature review to inform the RAM questionnaire items. We reviewed existing literature obtained through informal literature searches while preparing the study protocol. The titles and abstracts were scanned for relevance and screened against a set of inclusion and exclusion criteria [21]. We formally searched three online databases (i.e., PubMed, MEDLINE via Ovid, Google Scholar) to identify academic and grey literature. Lastly, we scanned the reference lists of included literature to obtain any additional materials. The full search strategy is detailed in our study protocol [21].

From the included texts, we extracted any quality indicators, metrics, measures, or processes to generate a candidate list. We assessed the relevance of candidate indicators for inclusion in our questionnaire by applying screening criteria (Table 1). Indicators must have satisfied all four criteria for inclusion.

Candidate quality indicators were then organized by the 18 "Priority Topics and Key Features for the Assessment of Competence in Care of the Elderly" (FM-COE Priority Topics) [29], which outline the bounds of best practices in caring for older adults [21]. For each Priority Topic, we grouped similar indicators to streamline our initial list and remove any duplicates. To finalize our RAM questionnaire items, we translated each indicator into a set of "if" and "then" statements and conducted an internal screening process to assess measurement feasibility and streamline the number of questionnaire items.

### Stage 2: Recruitment of technical expert panel

We described the procedures to identify and recruit prospective panelists previously [21]. Panelists included individuals with extensive knowledge about primary care for older persons, evidenced by at least five years of clinical practice experience or activity with older patients and/or at least two relevant academic publications. Prospective panelists reviewed our letter of information and completed our demographic survey in April 2023. We used findings from the demographic survey to tailor our sampling approach and ensure diverse representation.

**Table 1. Criteria for screening the relevance of candidate indicators.**

1. Does the indicator relate to <u>community-based</u> primary care? [a]

2. Does the indicator relate to the <u>role (scope of practice) of a family physician</u>? [b]

3. Is the indicator relevant to the <u>care of older adults</u>?

   - **Must be related** to the care of older adults **but not necessarily limited** to the care of older adults only.

4. Is the indicator <u>practice-based</u>? Is the indicator directly relevant to clinical activities/practice?

[a] Working definition of "community-based primary care": [26]

[b] Family physicians scope of practice definition: [27, 28]

## Stage 3: Achieving consensus in two RAM rounds

**Round #1 questionnaire.** Panelists completed the first questionnaire over a two-week period in May 2023 to rate candidate indicators on two criteria: appropriateness and importance. We defined and rationalized our modified RAM criteria in our study protocol [21]. Each indicator received ratings for each criterion on a 9-point Likert scale, ranging from 1 (extremely inappropriate; extremely not important) to 9 (extremely appropriate; extremely important). Participants had the opportunity to provide comments about the quality statements and their ratings within free-text comment boxes. All study data were collected and managed using the secure, web-based data capture platform, Research Electronic Data Capture (REDCap), hosted at McMaster University [30, 31].

As per RAM guidelines, indicators advanced to the second round if they achieved a median score between 7 and 9 for both criteria without disagreement, where disagreement was defined as three or more panelists rating in both extremes (i.e., 1–3 or 7–9) for either criterion [32]. Among indicators deemed eligible for inclusion in the second round, we rank-ordered the indicators to develop corresponding technical definitions efficiently and feasibly. We summed the median scores for each indicator on both criteria and computed interquartile ranges (IQRs). We sequenced indicators by those with the highest summed median score. For indicators with equal summed median scores, we prioritized those with the lowest summed IQR.

**Developing technical definitions.** We developed technical definitions for each endorsed indicator by suggesting approaches to operationalize the numerator ("then" portion of quality statement) and denominator ("if" portion). We anticipated prioritizing a subset of highly rated indicators to specify and present at our panel meetings. Within the previously determined meeting duration of two hours, we planned to allocate at least five minutes per indicator to gather feedback on each proposed technical definition. Based on the number of endorsed items from the first questionnaire, we planned to specify a threshold that would allow for a reasonable number of items to be discussed during our allotted meeting time (e.g., top 25% of the most highly rated endorsed indicators).

Health administrative data holdings at ICES in Ontario, Canada were referenced to identify relevant datasets and variables that expressed each indicator [33]. ICES is a central data repository for publicly funded administrative health services records in Ontario supporting population-based health research [34]. We identified, referenced, and modified pre-existing and validated indicators, specifications, classifications, procedures, outcomes, and derived cohorts, where possible [35–42]. We developed a standard template for each quality statement detailing its conceptual definition, interpretation, inclusion and exclusion criteria, data source (including relevant datasets and variable names), computation, and references. The proposed technical definitions were reviewed by two health administrative data experts (AC and AJ) in advance of the panel meeting.

**Synchronous panel meeting.** The purpose of the synchronous panel meeting was to review the endorsed indicators and gather feedback on their proposed technical definitions [21]. Panelists attended one of two virtual meetings (via Zoom) in June 2023. We were unable to identify a meeting time that satisfied the availability of all panelists, warranting two separate group meetings. Indicators were presented for discussion according to rank order; limited time and resources necessitated prioritizing a subset of indicators to present in each meeting.

**Round #2 questionnaire.** Following the group meetings, panelists rated the quality statements and corresponding technical definitions in a second questionnaire in June 2023. Panelists were only asked to rate indicators that were discussed during their panel meeting. We provided summary notes from the group meetings for panelists to consider while completing their ratings. As in the first questionnaire, panelists rated each indicator on both criteria and

could optionally provide free-text comments. Indicators comprising our final endorsed set included those rated between 7 and 9 on both criteria by more than 60% of panel members. Questionnaire responses were collected over a two-week period in June 2023.

## Stage 4: Analysis

Our data sources comprised the demographic survey and two RAM questionnaires. We summarized the demographic characteristics of panelists using descriptive statistics. Panelists reported their race/ethnicity, age, and gender within free-text comment boxes; we categorized participants' race and gender based on the responses provided verbatim and did not impose socially constructed groupings. After each questionnaire period concluded, we combined the de-identified judgements of panelists using statistical integration of Likert-scale ratings and conventional content analysis of open-ended responses [43]. We computed the median and IQR for each indicator on both criteria. We conducted the Wilcoxon matched-pairs signed-rank test to measure changes in consensus between rounds [44]. Qualitative data were used to revise quality statements and technical definitions. We shared individual feedback to panelists after each round, including summaries of their ratings relative to others. Statistical analyses were performed using Statistical Analysis Software (SAS) version 14 (Cary, North Carolina).

## Ethics and registration

Our study was approved by the Hamilton Integrated Research Ethics Board (#15545) and prospectively registered with ISRCTN (#17074347). All panelists provided written consent before data collection.

## Results

### Literature review

The search, screening, and extraction results of our literature review yielded 36 included texts and 500 candidate indicators (S1 Fig). We list and summarize the texts from which these indicators were obtained in S2 Table. Upon screening the extracted indicators, 282 were eligible for inclusion (n = 58 indicators obtained from the previously collected literature, n = 52 from the new literature search results, and n = 127 of those obtained from the reference lists of included literature).

We organized the 282 quality indicators by the 18 FM-COE Priority Topics and streamlined indicators within each Topic (S1 Fig), resulting in 178 items for internal screening. Most indicators aligned with "Medical Conditions" (n = 83) or "Appropriate Prescribing" (n = 73), while some did not align with any areas (n = 12). We grouped similar items and removed duplicates within each Priority Topic; this process eliminated 183 indicators. For example, nine items from different information sources pertained to the process of medication review and reconciliation and were collapsed into a single quality statement. The considerable degree of duplication in our candidate list suggested saturation, defined as information redundancy, across information sources [45].

We translated the eligible indicators into 178 quality statements using the "if"/"then" format. 112 statements were deemed "definitely not feasible" to measure within ICES data holdings via independent review in duplicate by health administrative data experts (APC and AJ); these statements were subsequently removed. Five quality statements were removed after a review for clinical accuracy by a physician (AG) and alignment with clinical practice guidelines (S1 Fig). Therefore, the literature review process resulted in 61 quality statements for inclusion in our first questionnaire, aligning with 10 FM-COE Priority Topics.

## Technical expert panel

We recruited 10 participants for our technical expert panel. All panelists reported involvement with health research for a median of 14 years (IQR: 8.75). Eight panelists held graduate degrees (i.e., master's degree or PhD) and nine were medical doctors (MDs); six physicians held "Care of the Elderly" Certificates of Added Competence. Clinicians had been in independent practice for a median of 22 years (IQR: 14). Panelists had a median age of 49 years (IQR: 11.25) and 70% were female (n = 7). Nine panelists were affiliated with institutions or organizations within Ontario; one resided and worked in another province. Seventy percent of panelists self-identified as white and 30% as East Asian, South Asian, or Southeast Asian. Eight panelists primarily worked or practiced in urban areas, one primarily in rural areas, and one in both urban and rural areas.

## Round #1 questionnaire

Across 61 questionnaire items, median scores ranged from 5.5 to 9.0 for appropriateness and 6.0 to 9.0 for importance (S3 Table). 55 indicators met our pre-specified threshold and advanced to the second round; six indicators were eliminated. Indicators that did not advance were rated poorly for not pertaining to the role/scope of family physicians or describing inappropriate clinical activities or timeframes.

Panelists' open-ended responses justified or clarified their ratings, suggested revisions to the wording of quality statements, or shared other reflections. For example, a participant rationalized their rating as follows: "*I rated importance lower because sometimes the info found in a CGA [Comprehensive Geriatric Assessment] would be included as part of the knowledge from continuity of care, not as a discrete assessment*" (Panelist #3, Indicator #2). Panelists confirmed activities that were "standard of care" (Panelist #10, Indicator #52) and criticized statements that were inaccurate or irrelevant. For example, a panelist questioned the role of family physicians in cardiac rehabilitation activities (Indicator #21),

> "*While it is important that cardiac rehab be offered, should this be done by cardiology? Maybe the PCP [Primary Care Provider] role is to verify that it has been offered.*"
>
> –Panelist #5

Participants often noted "*it depends*" (Panelist #4, Indicator #40) when rating statements, and clarified that the importance of performing any given clinical activity depended on the patient's severity, health status, goals of care, available resources, and other factors. Where relevant, panelists stated discrepancies between current practice guidelines and the given the clinical scenario or activity, such as: "*B12 is not in Canadian consensus guidelines*" (Panelist #2, Indicator #23). Some panelists shared references to jurisdictional and organizational guidelines, frameworks, policies, or research studies that facilitated or impeded the specified activities. For example, "*Offer all appropriate vaccines is consistent with the Canadian Frailty Network's AVOID framework*" (Panelist #5, Indicator #10). When diagnostic criteria or drugs were listed in the quality statement, participants frequently shared comments about adding or removing some of the specified items, such as: "*Consider adding sglt2 inhibitors*" (Panelist #2, Indicator #16).

## Technical definitions

From the 55 endorsed quality statements, it was necessary to develop technical definitions, an example of which is provided in S4 Table. We documented our efforts to operationalize each

quality statement and posed discussion questions within the technical definition workbook used to guide our discussion in the synchronous panel meeting. For example, if the frequency of a health service encounter or timeframe for follow-up was not specified within the quality statement, we noted these outstanding aspects as points of discussion.

We referenced four ICES datasets and three derived cohorts to draft technical definitions. The Primary Care Population (PCPOP) is an ICES dataset of eligible primary care patients in Ontario and contains information on demographics, primary care enrollment, healthcare utilization, specialist visits, and continuity of care. The Ontario Health Insurance Plan (OHIP) contains information on all billing claims submitted by physicians for consultations and procedures, including shadow billings for physicians practicing in non-fee-for-service models. The Ontario Laboratories Information System (OLIS) is an information repository containing lab test orders and results from hospitals, community labs and public health labs. The Ontario Drug Benefit (ODB) contains community-based, outpatient drug orders for products listed on the ODB formulary, notably for residents aged 65 years and older. We also referenced three ICES-derived cohorts that were previously developed using validated algorithms using diagnostic codes, hospitalization records, physician claims, and/or drug reimbursements for persons with dementia (DEMENTIA), chronic obstructive pulmonary disease (COPD), or congestive heart failure (CHF) [46–48].

We rank-ordered endorsed indicators from the first questionnaire to pragmatically determine a subset that would be reasonable to present in our allotted panel meeting times; it would not have been possible to discuss all 55 endorsed indicators during the predetermined two-hour timeframe. We considered what threshold of the most highly rated indicators would be feasible to present (e.g., top 10% = 6 indicators, top 25% = 14 indicators, etc.). We allotted 10 minutes to review the project objectives and facilitate introductions, 15 minutes to review ratings from the first questionnaire, and five minutes per indicator to gather feedback on the proposed technical definition–resulting in sufficient time to review 19 indicators (the top 35% most highly rated indicators) over 95 minutes (S5 Table; S2 Fig). In round 1, 14 indicators were rated most highly with summed median scores ranging from 18 to 16.5. 14 indicators achieved the next highest summed median score of 16; from these, 5 were prioritized based on having the lowest summed IQR.

## Synchronous panel meeting

We gathered feedback on the technical definitions during our synchronous panel meetings. Six panelists attended the first group meeting; four attended the second. The facilitator (RC) presented the technical definitions individually, and panelists provided feedback on the clinical scenario or activity. Panelists often commented on the frequency in which some physician fee codes were utilized, the diagnostic codes to specify clinical conditions, and medication groupings or classifications.

Through these discussions, we unanimously omitted four quality statements; two items were duplicates and two were not possible to accurately specify using administrative data. Panelists also suggested revising the language of eleven quality statements; these revised statements were incorporated into the second questionnaire and rated. Following the meetings, we finalized our technical definition workbook to reflect group feedback.

## Round #2 questionnaire

Panelists rated 15 quality statements and their corresponding technical definitions in the second questionnaire. Median scores ranged from 6.0 to 9.0 for appropriateness and 6.5 to 8.5 for importance (S6 Table). 12 indicators met our threshold for inclusion in the final indicator set

(Table 2); three indicators were eliminated. Drafted technical definitions for the endorsed indicators are available upon request.

Fig 1 illustrates the flow of candidate indicators across the study phases into our final set, broken down by FM-COE Priority Topics. The 12 endorsed quality statements expressed four Priority Topics: "Appropriate Prescribing" (n = 5), "Medical Conditions" (n = 4), "Cognitive Impairment" (n = 2), and "Driving Issues" (n = 1). The Wilcoxon matched-pairs signed-rank test achieved a P value of 0.0391, suggesting a significant difference in consensus scores between the two RAM rounds (S3 Fig).

Panelists' open-ended responses offered suggestions to the proposed technical definitions accompanying the indicators. Some comments affirmed the importance of indicators: "*A staple indicator in primary care*" (Panelist #10, Indicator #1). Others noted the limitations of technical definitions, such as physician fee codes used as a proxy measure of vaccination status. Participants noted regional differences inhibiting measurement in some jurisdictions (Indicators #7,8). Both supportive feedback and criticism was received for indicators that were re-worded during the synchronous panel meeting, including "*the revision is too broad and lacks context*" (Panelist #8, Indicator #13). Challenges of measuring some indicators were expressed: "*This indicator is appropriate, but as the discussion suggest[ed], will be very hard to measure/ assess so my rating goes down*" (Panelist #3, Indicator #2). Panelists also referenced guidelines,

**Table 2. Final endorsed quality indicator set by FM-COE Priority Topic.**

**Medical Conditions**

| | |
|---|---|
| 1 | IF an older primary care patient is eligible for the influenza vaccine, THEN the patient should be administered the vaccine annually. |
| 2 | IF an older primary care patient is not known to have already received a pneumococcal vaccine or if the patient received it more than 5 years ago (if before age 65 years), THEN a pneumococcal vaccine should be administered. |
| 3 | IF an older primary care patient has chronic obstructive pulmonary disease, THEN the primary care provider should recommend influenza and pneumococcal immunizations. |
| 4 | IF an older primary care patient presents with memory concerns, THEN the primary care provider should perform tests aligned with the 5th Canadian Consensus on Dementia.[a] |

**Appropriate Prescribing**

| | |
|---|---|
| 5 | IF an older primary care patient requires a new medication, THEN the primary care provider should not use benzodiazepines or other sedative-hypnotics as the first choice. |
| 6 | IF an older primary care patient requires a new medication, THEN the primary care provider should not prescribe a medication with strong anticholinergic effects if alternatives are available. |
| 7 | IF an older primary care patient requires medication, THEN the primary care provider should avoid prescribing potentially inappropriate medications (e.g., drugs from the Beers list). |
| 8 | IF an older primary care patient is prescribed medications from multiple providers, THEN the primary care provider should conduct a collaborative medication review (e.g., focus on evidence-based new drug prescriptions and prevention of polypharmacy). |
| 9 | IF an older primary care patient has congestive heart failure, THEN the primary care provider should order ACE inhibitors, ARBs, beta-blockers, or SGLT2 inhibitors. |

**Cognitive Impairment**

| | |
|---|---|
| 10 | IF an older primary care patient is diagnosed with dementia, THEN the primary care provider should provide dementia care management. |
| 11 | IF an older primary care patient is diagnosed with dementia, THEN the primary care provider should consider alternatives to antipsychotics as the first choice to treat. |

**Driving Issues**

| | |
|---|---|
| 12 | IF an older primary care patient receives a new diagnosis of dementia and is deemed unsafe to drive, THEN the primary care provider should report the patient to the Ministry of Transportation.[a] |

[a] Indicator also pertains to the Cognitive Impairment FM-COE Priority Topic

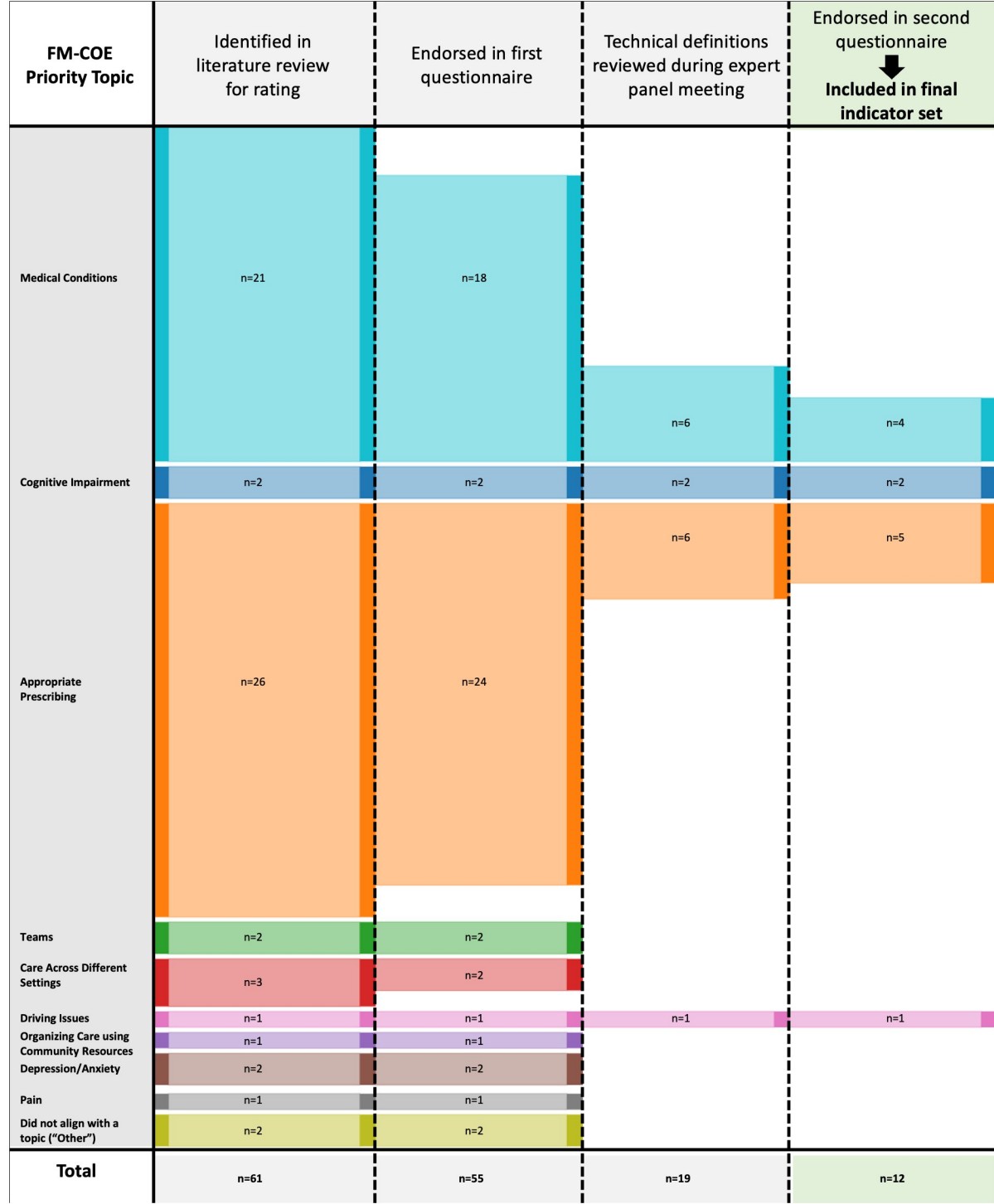

**Fig 1. Flow diagram of endorsed indicators, by FM-COE Priority Topic.** Legend: 61 indicators across 10 FM-COE Priority Topics were identified in the literature review for rating; 55 were endorsed in the first questionnaire. Of the 19 indicators reviewed in the panel meeting, 12 indicators across four Priority Topics were endorsed in the second questionnaire.

reports, research articles, and other documents to refine specifications, and suggested revisions to how some quality statements were stated. For example,

> *"To be future oriented, should the item read THEN "the PCP should perform tests aligned with the MOST CURRENT Canadian Consensus on Dementia". Presumably, the science/evidence will evolve and this language will prevent the indicator from becoming stale dated."*
>
> –Panelist #5, Indicator #9

## Discussion

We established consensus on a set of 12 practice-based process metrics and technical definitions to characterize quality of care for older primary care patients through the use of secondary health administrative data. The endorsed statements express four of 18 FM-COE Priority Topics and achieved increasingly greater ratings on "appropriateness" (round 1: 5.5 to 9.0; round 2: 6.0 to 9.0) and "importance" (round 1: 6.0 to 9.0; round 2: 6.5 to 8.5) in subsequent rounds. While the literature review revealed that many relevant indicators exist, our feasibility screening affirmed that most cannot be measured at scale to assess performance; we were unable to operationalize all candidate or endorsed indicators due to limitations within the available information sources. High ratings for the appropriateness and importance of indicators were achieved in both questionnaires, affirming the relevance of the included literature in which candidate indicators were extracted. Findings from this consensus study will support future work to test the endorsed indicators within population-based data holdings on their ability to distinguish medical practice of family physicians and quality of care for older patients.

While prior RAM studies have established quality standards or priorities to improve care for older patients in different care settings, we intended to develop a *measurable* indicator set. Limited efforts to date have yielded real world indicator translation supporting quality improvement [49, 50]. This process revealed that many indicators–those identified from the literature and endorsed through our consensus process–were concentrated on the biomedical aspects of caring for elderly patients–namely, the "Medical Conditions" (n = 4) and "Appropriate Prescribing" (n = 5) FM-COE Priority Topics. While developing measurable definitions for established indicators is not a required activity [23, 51], we intended to develop feasible indicator set to reference readily collected data to assess performance; measurement is essential to inform and support quality improvement. Whereas some Priority Topics lend themselves well to measurement using administrative data (e.g., by specifying diagnostic codes, drug identification numbers), others were not possible to specify (e.g., "Communication," "Decision making and capacity," "Family and informal care supports"). More work can be done to develop approaches to measure other facets of elderly-focused care.

Further, our panelists with demonstrated clinical and research expertise likely prioritized indicators differently than older adults or informal caregivers would have. Patients are increasingly involved in quality improvement initiatives and their first-hand experiences and inputs into problem-solving have demonstrated value in redesigning care aligned with patient priorities [52–54]. For example, indicators relevant to "organizing care using community resources," a FM-COE Priority Topic, were identified in our literature review but not endorsed. Therefore, by not including patients or caregivers on our expert panel, the endorsed indicators may only constitute some of the primary care activities that are important or meaningful to patients. Through a separate but complementary study, we will engage older adults to compare and establish patient-important indicators. This approach will extend how we conceptualize and

assess quality primary care for older adults, and maximize patient perspectives and feedback in an open-ended format, rather than limiting their viewpoints to the constraints of feasible secondary data.

Measurable quality indicators are limited by information sources available, and technical definitions–which are necessary to operationalize indicators–pertain to particular settings, thereby, affecting their generalizability [21]. There is substantial diversity in how clinical scenarios and activities are defined in different healthcare contexts and reported in administrative data sources, which prohibits the development of universal technical definitions. In an effort to establish measurable indicators to support future work, technical definitions developed through this process are contextualized to community-based primary care settings in Ontario, Canada, and relevant information sources. While the specific data set and variable names of our endorsed indicators may not be directly apply to other information sources, they can be adapted to the local context if studied in other settings [55].

Given current and projected demands on the geriatric healthcare workforce [56, 57], efforts are needed to improve the quality of medical care delivered to complex older patients [58]. This study focused on testing *whether* technical definitions can be established that define quality primary care for this vulnerable population. Quality measures do not function well on the individual (in this case, provider) level since other factors (e.g., setting, resources) may impact processes or resulting health outcomes. Therefore, we do not suggest that these indicators should be implemented in practice (such as to generate individual-level reports); rather, our findings may lead to education and awareness about approaches to measuring practice-based primary care encounters and for quality improvement. Ultimately, the use of indicators must be driven by the professional community.

## Limitations

The technical expert panel was composed mostly of primary care clinicians and researchers who worked or practiced in urban areas and lacked representation of some ethnic minorities and experts located outside Ontario. Although we recruited fewer panelists than anticipated [21], all participants completed both questionnaires and actively contributed in the synchronous panel meetings. Due to the volume of indicators identified in our literature review, we restricted items for rating in the first questionnaire to those that were potentially feasible to measure using administrative data. Our focus on developing a *measurable* indicator set excluded many items that were not currently collected/available in data holdings; future work could examine excluded indicators as new measures become available or within other information sources. For example, we could not specify physicians' referrals to social services or resources (e.g., adult day programs) or patients' symptoms (e.g., insomnia, agitation, delirium), and we used physician fee codes as a proxy for some clinical activities (e.g., immunizations).

We only presented a subset of endorsed indicators from the first questionnaire and their technical definitions in the group panel meetings. Although excluding indicators for pragmatic reasons (i.e., the predetermined two-hour panel meeting length) may have biased our findings, our study demonstrated the utility of using a consensus process to derive a measurable indicator set for those rated most highly. We prioritized indicators for discussion in the panel meeting by rank-ordering endorsed indicators from round 1 with the highest summed median scores and, for those tied, the lowest summed IQR. We did not initially set out to establish a comprehensive indicator set; rather, we aimed to determine whether it was possible to establish consensus on measurable indicators by referencing administrative data holdings [21]. Lastly, we did not test our technical specifications for the endorsed indicators; these definitions

will be operationalized and refined in future work. If a useful profile can be created, we can re-engage panelists to solicit feedback on the outstanding indicators (S5 Table).

## Conclusion

We produced a measurable set of quality indicators that will support work to examine primary care provision for older adults using health administrative data. While not comprehensive of all 18 FM-COE Priority Topics, the endorsed indicators provide a framework to characterize practice- and population-level encounters of family physicians delivering care to older patients and will offer insights into the outcomes of their care provision. The technical definitions established through this study offer a case to trial their measurement within the context of population-based data sources in Ontario, Canada. There is work to be done to understand the feasibility of operationalizing the drafted specifications. Ultimately, these efforts may identify areas where high quality care for elderly patients is consistently provided and systematic challenges in delivering elderly-focused care. Measuring primary care quality for older adults using these metrics is a starting point to determine opportunities for resources, education, incentives, interventions, and policies to support quality improvement.

## Supporting information

**S1 Fig. Literature search, organization, and screening results.** Legend: Steps 1 to 4 outline the literature search results of 282 candidate indicators. Steps 5 to 7 illustrate how the candidate indicators were organized by the FM-COE Priority Topics and streamlined into 178 quality statements. Steps 8 and 9 display the results of internal feasibility screening and clinical review, resulting in 61 quality statements for inclusion in the first questionnaire.
(TIFF)

**S2 Fig. Distribution of summed median scores and IQRs for indicators that advanced from round 1.** Legend: 19 indicators whose summed median scores were rated most highly were prioritized for discussion in the group panel meeting. 14 indicators had equal summed median scores and were rank-ordered by the lowest summed IQR for prioritization.
(TIFF)

**S3 Fig. Distribution of Wilcoxon scores (matched-pairs signed-rank test).** Legend: The p-value obtained by the Wilcoxon matched-pairs signed-rank test suggests a significant difference in consensus scores between the two RAM rounds.
(TIFF)

**S1 Table. Recommendations for the conducting and REporting of DElphi studies (CREDES) checklist.**
(DOCX)

**S2 Table. Summary of included literature.**
(DOCX)

**S3 Table. Summary of round 1 ratings.**
(DOCX)

**S4 Table. Example of a technical definition for an endorsed quality statement.**
(DOCX)

**S5 Table. Prioritization of endorsed items from round 1 to round 2.**
(DOCX)

**S6 Table. Summary of round 2 ratings.**
(DOCX)

## Author Contributions

**Conceptualization:** Rebecca H. Correia, Aaron Jones, Meredith Vanstone, Henry Yu-Hin Siu, Andrew P. Costa.

**Formal analysis:** Rebecca H. Correia.

**Investigation:** Rebecca H. Correia, Darly Dash, Aaron Jones, Komal Aryal, Aquila Gopaul.

**Methodology:** Rebecca H. Correia, Aaron Jones, Meredith Vanstone, Henry Yu-Hin Siu, Andrew P. Costa.

**Project administration:** Rebecca H. Correia.

**Supervision:** Andrew P. Costa.

**Validation:** Darly Dash, Komal Aryal, Aquila Gopaul.

**Writing – original draft:** Rebecca H. Correia.

**Writing – review & editing:** Darly Dash, Aaron Jones, Meredith Vanstone, Komal Aryal, Henry Yu-Hin Siu, Aquila Gopaul, Andrew P. Costa.

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
