## [Decision Letter · Decision Letter 0]

9 Nov 2023

PONE-D-23-32654Primary care quality for older adults: practice-based quality measures derived from a RAND/UCLA appropriateness method studyPLOS ONE

Dear Dr. Correia,

Thank you for submitting your manuscript to PLOS ONE. After careful consideration, we feel that it has merit but does not fully meet PLOS ONE’s publication criteria as it currently stands. Therefore, we invite you to submit a revised version of the manuscript that addresses the points raised during the review process.

Based on the reviewers' feedback, which includes three different decisions, I kindly request that you consider making **significant revisions** to your work. I want to be transparent and inform you that if any of the reviewers reject your revised version, I will promptly reject it as well.

We look forward to receiving your revised manuscript.

Kind regards,

Mohammad Sadegh Taghizadeh

Academic Editor

PLOS ONE

Journal Requirements:

2. Please amend your list of authors on the manuscript to ensure that each author is linked to an affiliation. Authors’ affiliations should reflect the institution where the work was done (if authors moved subsequently, you can also list the new affiliation stating “current affiliation:….” as necessary).

Reviewers' comments:

Reviewer's Responses to Questions

**Comments to the Author**

1. Is the manuscript technically sound, and do the data support the conclusions?

Reviewer #1: Partly

Reviewer #2: Yes

Reviewer #3: Yes

2. Has the statistical analysis been performed appropriately and rigorously? 

Reviewer #1: I Don't Know

Reviewer #2: Yes

Reviewer #3: Yes

3. Have the authors made all data underlying the findings in their manuscript fully available?

Reviewer #1: Yes

Reviewer #2: Yes

Reviewer #3: Yes

4. Is the manuscript presented in an intelligible fashion and written in standard English?

Reviewer #1: Yes

Reviewer #2: Yes

Reviewer #3: Yes

5. Review Comments to the Author

Reviewer #1: This is generally a well-written manuscript with general concern as immediately below ("line 57 and generally") towards suitability for review and publication in PLOS ONE. Other edits are thereafter detailed.

--line 57 and generally: since this is an effort to define practice-based quality measures for geriatric care (through consensus of a technical expert panel), I think this would be best reviewed and considered by a geriatric and/or primary care journal, perhaps even a Canada-specific journal (Canadian Family Physician?) given the potential lack of generalizability across health service systems and consensus of only 10 individuals from a relatively limited geographic area (is this "technically sound"?). or even BMJ Open where the protocol paper was published?

--line 43, lines 306-307: em dashes should be here without spaces

--line 71: why an informal literature search vs. systematic or scoping review?

--table 1:  I understand “inclusive but not necessarily exclusive” but this statement may benefit from more straightforward wording? “must be related to the care of older adults but not necessarily limited to the care of older adults only”?

--line 201: since scores were skewed, can scores be presented as a scatter plot with a line at 35% to clearly see the raw score difference for the top 19 vs. other items?

table 2:

--item 2: a new medication for what?

--item 4: what is dementia care management?

--are items 2 and 3 not included in item 5?

--is item 6 not included in item 4? this seems specific to Canada?

--why are the items listed in this order? item 8 seems to logically come before item 4?

--the differences between item 9 and items 1 and 10 seem minimal…

--item 11: understand perhaps hard to “measure” (and acknowledged lines 321-322) but for safe clinical interpretation, feel the following clarifications are necessary — ACE/ARB/ARNI to clarify (patients shouldn’t be more than one of these), an appropriate beta-blocker (carvedilol, long-acting metoprolol, or bisoprolol), and SGLT2 inhibitor, all pending clinical appropriateness and renal function (i.e., will have to be more carefully considered or avoided with allergies, advanced renal disease, and/or hypotension)

--item 12: is this not included in item 4?

--again, why are items listed in this order? would personally prefer some grouping (e.g., vaccines, dementia, medication management)

--line 296: we does not need capitalized

--line 311: measurement does not need capitalized

--lines 317-319: this sentence seems mis-edited? established or endorsed that define primary care quality?

--line 322: rather does not need capitalized

--line 336: future does not need capitalized

--lines 344-345: I understand these as practical limitation. almost everything is limited by time and resources but this is not a justification to publish incomplete science. stating it here is not necessary and detracts from the meaningful work that is otherwise presented.

--line 348: rather does not need capitlized

--line 350: these does not need capitalized

Reviewer #2: The purpose of the study is to establish consensus on practice-based metrics that characterize quality of community-based primary healthcare for older primary care patients. The quality indicators should be accessible using health administrative data.

There is a need for patient-centered appropriate and important quality indicators.

The current project basis its methods on the RAND/UCAL Appropriateness Methods (RAM). Referred to as an established methods. it was adopted to fit in terms of establishing indicators for older patients. Apart from establishing a set of quality indicators, the paper also present data on the technical definitions of the respective quality domains assessed.

Abstract

The abstract is comprehensive and appropriately reflects the study design and findings.

Introduction

The introduction covers rel3evant areas and references.

Major challenges to using administrative data include difficulty getting access to relevant/sought after data. Much adm data is gathered for billing and monitoring productivity. The manuscript discusses the need to select indicators for which administrative data is available. However, the manuscript should recognize, and discuss, in the intro the challenges of using administrative data, the need to have provider buy-ins, that these identifiers reflect quality of care. Moreover, the manuscript should refer to other large administrative databases that exist in terms of assessing healthcare provisions for elderly persons. For example, in the US Health grade. The introduction largely fails to review the pros and cons of using administrative data only without engaging the patient perspective.

Methods

State 2 is well described. it is a strength that Priority Topics and Key Features for assessment of Competence in Care of Elderly (FM-COE Priority Topics were used to organize quality indicators.

Stage 2What was the response/acceptance rate? That is, out of xx providers approached, how many accepted.

Stage 3 Clarify line 108. If th median score was betwen 7 and 9, how could an extreme rating be defined as (i.e. xx or 7-9)?

Developing Technical Definition: what is the dealy between clinical/adm data is collected and processed, and assessible in ICES? How reevant are proposed datasets to determine quality indicators in real-life - limiting the delay in feedback as to make data relevant for QI efforts in the clinic?

Synchronous panel meeting: Did the second meeting, when experts reviewed and scored proposed quality indicators also done virtually in a panel format?

Results

Fig 1. Expand in text what the figure is supposed to represent - what is the conclusion?

Fig 2, similar to above, expand the explanatory text in the manuscript.

Technical panel, line 190 and ff. Move descriptive part to Methods, where Provider characteristics are presented.

Consider reducing number of tables/figures.

Discussion

Statement that consensus was established - line 292. Provide data support for the statement. Moreover, discuss relatively low median scores for both Appropriateness and Importance (5-7). Sounds like these quality indicators are not rated very high by neither providers, nor technical experts in terms of Appropriateness and Importance.

The manuscript contains a substantial amount of data, specifically in a rather large number of attachments. This data is only limited referred to in the result section. moreover, even less discussed in the discussion section. Rather, generic comments are provided, e. g., line 307 and 308: "-were concentrated on biomedical aspects of caring for elderly patients - namely, the "Medical Conditions". This is well known, lack of functional quality indicators, survival data, and patient-perspectives are major limitations in current quality of care indicator data. as in this article, much data is process rather than outcome focused.

The Discussion section is very limited, does not refer to the greater debate of relevance, and hardly et al refer to other studies of relevance. Furthermore, the authors should discuss the relevance of using only expert-derived quality indicators in an age of patient-centered primary care. Lack of including patients or caregivers are briefly touched on in Limitation (ll 340-343. However, referring to another study to be, does not address this major shortcomings. Even when exploring how to use administrative data. such data can be generated for patients and caretakers as well.

Reviewer #3: thank you for reviewing this interesting paper. The study is well conducted and provides an interesting view on how to characterize quality of care for older primary care patients on population level.

Some questions and minor comments could enhance the readability of the paper.

Introduction

Please clarify what you mean with ‘practice-based process metrics’An operationalization of quality of care Is not described, please elaborate how this concept is viewed by the authors.

Methods

The steps are easy to follow. Hence, please clarify the following aspects.1. Table 1 point 4 “practice-based orientation.

Results

Round 1 questionnaire please describe the six indicators that were eliminatedMany quotes from the panelists are described, what was done with this information?How many panelists were participating during each session?

Discussion

Please elaborate more why and how the selection was done to reduce the number of indicators – only time limitation is somewhat vague.Please elaborate on the generalizability of the available administrative data

6. PLOS authors have the option to publish the peer review history of their article (what does this mean?). If published, this will include your full peer review and any attached files.

Reviewer #1: No

Reviewer #2: **Yes: **Bengt B. Arnetz, MD, PhD

Reviewer #3: **Yes: **dr. Nienke Bleijenberg

---

## [Author Response · Author response to Decision Letter 0]

6 Dec 2023

Reviewer #1:

1. This is generally a well-written manuscript with general concern as immediately below ("line 57 and generally") towards suitability for review and publication in PLOS ONE. Other edits are thereafter detailed.

--line 57 and generally: since this is an effort to define practice-based quality measures for geriatric care (through consensus of a technical expert panel), I think this would be best reviewed and considered by a geriatric and/or primary care journal, perhaps even a Canada-specific journal (Canadian Family Physician?) given the potential lack of generalizability across health service systems and consensus of only 10 individuals from a relatively limited geographic area (is this "technically sound"?). or even BMJ Open where the protocol paper was published?

Thank you for this feedback. We intend to publish in PLOS ONE because our findings and the implications of our work are informative and relevant to multidisciplinary audiences – for example, primary care professionals directly engaged in caring for older adults; medical educators developing training/assessment opportunities related to family medicine, care of the elderly, and geriatrics; and health system (primary care) planners. We value PLOS ONE’s mission to make research available and discoverable for all audiences by reducing barriers for readers (e.g., financial). PLOS ONE is a widely disseminated journal and not discipline-specific (e.g., catering to both primary care and geriatrics audiences), facilitating the reach of our work.

We appreciate your comments surrounding the generalizability of our results and consultation of a pan-Canadian expert panel. We believe the novelty of our work is in establishing measurable quality indicators using the case of population-based data holdings at ICES in Ontario, Canada. In contrast to much quality improvement work that does not provide measurable indicators, we advance the methodological literature by demonstrating a study where expert panelists not only endorsed an indicator set, but also their corresponding technical definitions in reference to administrative data holdings. While we recognize these indicators were established within a particular setting/context, our work examines the feasibility of establishing measurable definitions within this case that can then be adapted elsewhere. There is much diversity (provincially, nationally, internationally) in how health administrative data are collected and made available to researchers, which means the specific technical definitions that we developed (e.g., dataset names, variable names, timings) will only apply within the ICES context. In order to be measurable, we had to select a case (i.e., ICES) to reference administrative data holdings to draft our technical definitions. However, there is a high probability that the operational definitions we develop can be adapted to other contexts for meaningful research and evaluation.

Based on your feedback and suggestions from other reviewers, we have since elaborated on the generalizability of our study in the Discussion.

Lines 353-361: Measurable quality indicators are limited by information sources available and technical definitions – which are necessary to operationalize indicators – pertain to particular settings, thereby, affecting their generalizability [21]. There is substantial diversity in how clinical scenarios and activities are defined in different healthcare contexts and reported in administrative data sources, which prohibits the development of universal technical definitions. In an effort to establish measurable indicators to support future work, technical definitions developed through this process are contextualized to community-based primary care settings in Ontario, Canada, and relevant information sources. While the specific data set and variable names of our endorsed indicators may not be directly apply to other information sources, they can be adapted to the local context if studied in other settings [55].

Finally, in our Conclusion, we state how our findings are informative beyond the case/context in which our study was conducted. 

Lines 393-403: We produced a measurable set of quality indicators that will support work to examine primary care provision for older adults using health administrative data. While not comprehensive of all 18 FM-COE Priority Topics, the endorsed indicators provide a framework to characterize practice- and population-level encounters of family physicians delivering care to older patients and will offer insights into the outcomes of their care provision. The technical definitions established through this study offer a case to trial their measurement within the context population-based data sources in Ontario, Canada. There is work to be done to understand the feasibility of operationalizing the drafted specifications. Ultimately, these efforts may identify areas where high quality care for elderly patients is consistently provided and systematic challenges in delivering elderly-focused care. Measuring primary care quality for older adults using these metrics is a starting point to determine opportunities for resources, education, incentives, interventions, and policies to support quality improvement.

2. --line 43, lines 306-307: em dashes should be here without spaces

Thanks for your comment. In our Word processing system, we do not see spaces in the em dashes. 

3. --line 71: why an informal literature search vs. systematic or scoping review?

Thanks for this question. The purpose of our literature search was to inform the initial set of candidate quality indicators and understand the state of indicator development, methodological quality, and current gaps in indicator availability. The RAND/UCLA Appropriateness Method (RAM) User’s Manual does not require that literature reviews for appropriateness studies be systematic. 

Therefore, given our objectives to establish a measurable indicator set (not necessarily a comprehensive set) and RAM methodological guidance, we chose to conduct an informal literature search. However, as detailed in our published study protocol, we provide a full search strategy including a review of readily available materials, a formal search of three online databases, and a review of the references of included texts. Further, recognizing that the literature review may have missed some potentially important candidate indicators, we provided an opportunity for panelists to elect new indicators during the panel group meeting (as is commonly done in RAM studies).

The following excerpts from the RAM manual describe why systematic or scoping reviews may not be warranted for RAM studies: 

• “The objectives of a literature review developed for an appropriateness study are somewhat different: a Cochrane review is generally limited to scientific evidence from randomised controlled trials or similarly methodologically rigorous research, whereas an appropriateness review includes the best available evidence, which may not always meet Cochrane standards.” (Page 10)

• “Cochrane reviewers use a precise, standardised methodology to review potential articles, with strict inclusion and exclusion criteria to assure that only evidence from well-conducted randomised controlled trials is included. […] Literature reviews for appropriateness studies are typically less strict in their inclusion criteria, as the objective is to produce a synthesis of all the information available on a particular topic; where evidence from controlled trials is lacking, they may well include lower quality evidence from, for example, cohort studies or case series.” (Page 10)

4. --table 1: I understand “inclusive but not necessarily exclusive” but this statement may benefit from more straightforward wording? “must be related to the care of older adults but not necessarily limited to the care of older adults only”?

Thank you for this suggested phrase. We have incorporated this language into our main text. 

Table 1: Must be related to the care of older adults but not necessarily limited to the care of older adults only.

5. --line 201: since scores were skewed, can scores be presented as a scatter plot with a line at 35% to clearly see the raw score difference for the top 19 vs. other items?

Thanks for this suggestion. We generated the scatterplot below to illustrate the summed median scores (for importance and appropriateness) for indicators that met our threshold to advance from round 1 (n=55). As shown in this figure and described in our Results, we prioritized indicators for discussion in the group panel meeting in two ways. First, we rank-ordered indicators by those with the greatest summed median scores. Next, for those with an equal summed median score, we sequenced them by those with the smallest interquartile range [IQR] to reflect less dispersion in ratings. Given the allotted time to facilitate our group panel meetings, we prioritized the top 19 (35%) most highly rated indicators to present/discuss with panelists. 

If this figure is helpful to illustrate the raw scores of the top 19 items, we can consider adding it as a supplement. 

6. table 2:

--item 2: a new medication for what?

--item 4: what is dementia care management?

--are items 2 and 3 not included in item 5?

--is item 6 not included in item 4? this seems specific to Canada?

--why are the items listed in this order? item 8 seems to logically come before item 4?

--the differences between item 9 and items 1 and 10 seem minimal…

--item 11: understand perhaps hard to “measure” (and acknowledged lines 321-322) but for safe clinical interpretation, feel the following clarifications are necessary — ACE/ARB/ARNI to clarify (patients shouldn’t be more than one of these), an appropriate beta-blocker (carvedilol, long-acting metoprolol, or bisoprolol), and SGLT2 inhibitor, all pending clinical appropriateness and renal function (i.e., will have to be more carefully considered or avoided with allergies, advanced renal disease, and/or hypotension)

--item 12: is this not included in item 4?

--again, why are items listed in this order? would personally prefer some grouping (e.g., vaccines, dementia, medication management)

Thanks for your feedback on our endorsed indicator set presented in Table 2. The open-ended responses we received in our second questionnaire (summarized from Lines 311 to 336) suggest – in line with your comments – that there is still work to be done to finalize these quality statements. As you noted above, panelists suggested for some statements to be rephrased, additional resources should be referenced in their specifications, and other details should be included to finalize this indicator set. We appreciate this feedback and will continue to refine these statements in consultation with clinicians as we engage in future work to operationalize and test these indicators within administrative data holdings at ICES. For the purpose of this study/manuscript, the 12 quality statements – as written – were endorsed by our expert panel through this RAM process. 

The following two excerpts from our main text describe that there is still work to be done to finalize these statements: 

Lines 388-391: Lastly, we did not test our technical specifications for the endorsed indicators; these definitions will be operationalized and refined in future work. If a useful profile can be created, we can re-engage panelists to solicit feedback on the outstanding indicators (S6 Table).

Lines 397-399: The technical definitions established through this study offer a case to trial their measurement within the context population-based data sources in Ontario, Canada. There is work to be done to understand the feasibility of operationalizing the drafted specifications.

Lastly, we agree your suggestion to better organize the statements in this table. We have since added headings corresponding to the FM-COE Priority Topics in Table 2. 

Table 2. Final endorsed quality indicator set by FM-COE Priority Topic

Medical Conditions

1 IF an older primary care patient is eligible for the influenza vaccine, THEN the patient should be administered the vaccine annually.

2 IF an older primary care patient presents with memory concerns, THEN the primary care provider should perform tests aligned with the 5th Canadian Consensus on Dementia.

3 IF an older primary care patient has chronic obstructive pulmonary disease, THEN the primary care provider should recommend influenza and pneumococcal immunizations.

4 IF an older primary care patient is not known to have already received a pneumococcal vaccine or if the patient received it more than 5 years ago (if before age 65 years), THEN a pneumococcal vaccine should be administered.

Appropriate Prescribing 

5 IF an older primary care patient requires a new medication, THEN the primary care provider should not use benzodiazepines or other sedative-hypnotics as the first choice.

6 IF an older primary care patient requires a new medication, THEN the primary care provider should not prescribe a medication with strong anticholinergic effects if alternatives are available.

7 IF an older primary care patient requires medication, THEN the primary care provider should avoid prescribing potentially inappropriate medications (e.g., drugs from the Beers list).

8 IF an older primary care patient is prescribed medications from multiple providers, THEN the primary care provider should conduct a collaborative medication review (e.g., focus on evidence-based new drug prescriptions and prevention of polypharmacy).

9 IF an older primary care patient has congestive heart failure, THEN the primary care provider should order ACE inhibitors, ARBs, beta-blockers, or SGLT2 inhibitors.

Cognitive Impairment 

10 IF an older primary care patient is diagnosed with dementia, THEN the primary care provider should provide dementia care management.

11 IF an older primary care patient is diagnosed with dementia, THEN the primary care provider should consider alternatives to antipsychotics as the first choice to treat.

Driving Issues

12 IF an older primary care patient receives a new diagnosis of dementia and is deemed unsafe to drive, THEN the primary care provider should report the patient to the Ministry of Transportation.

7. --line 296: we does not need capitalized

Thanks for catching this grammatical error; we have revised accordingly.

8. --line 311: measurement does not need capitalized

Thanks for catching this grammatical error; we have revised accordingly.

9. --lines 317-319: this sentence seems mis-edited? established or endorsed that define primary care quality?

Thanks for catching this grammatical error; we have removed “for endorsed” from the main text. 

Lines 394-396: This study focused on testing whether technical definitions can be established that define quality primary care quality for this vulnerable population.

10. --line 322: rather does not need capitalized

Thanks for catching this grammatical error; we have revised accordingly.

11. --line 336: future does not need capitalized

Thanks for catching this grammatical error; we have revised accordingly.

12. --lines 344-345: I understand these as practical limitation. almost everything is limited by time and resources but this is not a justification to publish incomplete science. stating it here is not necessary and detracts from the meaningful work that is otherwise presented.

Thank you for recognizing this practical limitation and for your suggestions to improve our explanation. When we originally specified our thresholds for advancement between rounds, we were not expecting so many indicators to be rated so highly. This resulted in having far too many indicators (n=55) to present/discuss in our previously scheduled panel group meetings, which were allotted 2 hours in our original letter of information to panelists. We detailed this in the following excerpt: 

Lines 249-258: From the 55 endorsed quality statements, it was necessary to develop technical definitions, an example of which is provided in S4 Table. We rank-ordered the endorsed indicators from the first questionnaire to pragmatically determine a subset that would be reasonable to present in our allotted panel meeting times; it would not have been possible to discuss all 55 endorsed indicators during the predetermined two-hour timeframe. We considered what threshold of the most highly rated indicators would be feasible to present (e.g., top 10% = 6 indicators, top 25% = 14 indicators, etc.). We allotted 10 minutes to review the project objectives and facilitate introductions, 15 minutes to review ratings from the first questionnaire, and five minutes per indicator to gather feedback on the proposed technical definition – resulting in sufficient time to present 19 indicators (the top 35% most highly rated indicators) over 95 minutes. 

Rank-ordering indicators based on those rated most highly allowed us to seek feedback on as many endorsed indicators as possible in our allotted panel group meetings. As per your suggestion, we have removed our comments about time and resources influencing this methodological decision from our Limitations. 

Lines 383-388: We only presented a subset of endorsed indicators from the first questionnaire and their technical definitions in the group panel meetings. Although excluding indicators for pragmatic reasons may have biased our findings, our study demonstrated the utility of using a consensus process to derive a measurable indicator set for those rated most highly. We did not initially set out to establish a comprehensive indicator set; rather, we aimed to determine whether it was possible to establish consensus on measurable indicators by referencing administrative data holdings [21].

Concluding the Limitations with the following sentence reinforces that we plan to seek feedback on the other endorsed indicators (that were not prioritized for our panel group meeting) in future work: 

Lines 390-391: If a useful profile can be created, we can re-engage panelists to solicit feedback on the outstanding indicators (S6 Table).

13. --line 348: rather does not need capitlized

Thanks for catching this grammatical error; we have revised accordingly.

14. --line 350: these does not need capitalized

Thanks for catching this grammatical error; we have revised accordingly.

Reviewer #2:

The purpose of the study is to establish consensus on practice-based metrics that characterize quality of community-based primary healthcare for older primary care patients. The quality indicators should be accessible using health administrative data. There is a need for patient-centered appropriate and important quality indicators. The current project basis its methods on the RAND/UCAL Appropriateness Methods (RAM). Referred to as an established methods. it was adopted to fit in terms of establishing indicators for older patients. Apart from establishing a set of quality indicators, the paper also present data on the technical definitions of the respective quality domains assessed.

Abstract

The abstract is comprehensive and appropriately reflects the study design and findings.

Introduction

1. The introduction covers relevant areas and references.

Major challenges to using administrative data include difficulty getting access to relevant/sought after data. Much adm data is gathered for billing and monitoring productivity. The manuscript discusses the need to select indicators for which administrative data is available. However, the manuscript should recognize, and discuss, in the intro the challenges of using administrative data, the need to have provider buy-ins, that these identifiers reflect quality of care. Moreover, the manuscript should refer to other large administrative databases that exist in terms of assessing healthcare provisions for elderly persons. For example, in the US Health grade. The introduction largely fails to review the pros and cons of using administrative data only without engaging the patient perspective.

Thank you for this feedback. We acknowledge that we describe the limitations and challenges of relying on secondary (administrative) data in our published study protocol, but did not detail this sufficiently in our original submission. We have since added the following phrase in our Introduction and reference to our study protocol:

Lines 72-74: Despite the limitations of using secondary data to make inferences about quality [21], indicators developed without reference to information sources or those that are not feasible to operationalize within a particular context are limited in their utility and capacity to promote change.

The following excerpts from our published study protocol discuss the limitations you have noted. 

Excerpt from published study protocol (Correia, R. H., Siu, H. Y. H., Vanstone, M., Jones, A., Gopaul, A., & Costa, A. P. (2023). Development of practice-based quality indicators for the primary care of older adults: a RAND/UCLA Appropriateness Method study protocol. BMJ Open, 13(9), e072232): 

“One approach to measuring healthcare quality is utilising secondary (health administrative) data;20 21 although examining indicators in practice are limited by the information sources available.”

“Additionally, our focus on practice-based quality measures may exclude some aspects of primary care that are not captured in administrative data. For example, characteristics of primary care encounters such as time spent with individual patients or engagement with informal caregivers may be clinically meaningful and important to patients, but are not available in billing data. Similarly, some components of the ‘Priority Topics and Key Features for the Assessment of Competence in Care of the Elderly’ may be over-represented or under-represented in our final indicator set based on their availability within health administrative data. For example, we anticipate it will be highly feasible to operationalise indicators related to ‘medical conditions’ because diagnostic codes and billing data for physician services are readily available in administrative data. However, indicators related to ‘goals of care’ may not have objective measures to specify within data holdings. Therefore, the practice-based processes included in our final indicator set may only constitute some of the primary care activities delivered to older patients.” 

In response to your feedback and comments from another reviewer, we have since added a paragraph in our Discussion about the limitations of using administrative data to measure and make inferences about quality of care. We agree with your suggestion about the need to recognize and discuss these limitations/challenges. 

Lines 353-361: Measurable quality indicators are limited by information sources available and technical definitions – which are necessary to operationalize indicators – pertain to particular settings, thereby, affecting their generalizability [21]. There is substantial diversity in how clinical scenarios and activities are defined in different healthcare contexts and reported in administrative data sources, which prohibits the development of universal technical definitions. In an effort to establish measurable indicators to support future work, technical definitions developed through this process are contextualized to community-based primary care settings in Ontario, Canada, and relevant information sources. While the specific data set and variable names of our endorsed indicators may not be directly apply to other information sources, they can be adapted to the local context if studied in other settings [52].

Further, we now discuss the need to have provider buy-in in our Discussion. We believe the value of these indicators is to identify educational and practice gaps on the population-level to inform quality improvement initiatives (e.g., geriatric-focused continuing education for family physicians), rather than to assess individual providers on the “performance” of these indicators. 

Lines 365-370: Quality measures do not function well on the individual (in this case, provider) level since other factors (e.g., setting, resources) may impact processes or resulting health outcomes. Therefore, we do not suggest that these indicators should be implemented in practice (such as to generate individual-level reports); rather, our findings may lead to education and awareness about approaches to measuring practice-based primary care encounters and for quality improvement. Ultimately, the use of indicators must be driven by the professional community.

Methods

State 2 is well described. it is a strength that Priority Topics and Key Features for assessment of Competence in Care of Elderly (FM-COE Priority Topics were used to organize quality indicators.

2. Stage 2 What was the response/acceptance rate? That is, out of xx providers approached, how many accepted.

Thanks for this inquiry. In total, we approached 20 prospective panelists with an intended panel size of 12 to 15 persons (specified in our published study protocol). Of those we approached, 12 persons agreed to participate and 8 declined. 

However, two participants who agreed to participate did not respond to our repeated attempts of distributing the consent letter and demographic survey. We made six attempts of contacting these two individuals but did not receive a response. Therefore, they were not contacted when it came time to distribute the first questionnaire – bringing our final/total panel size was 10 participants. 

3. Stage 3 Clarify line 108. If th median score was betwen 7 and 9, how could an extreme rating be defined as (i.e. xx or 7-9)?

We acknowledge that is sentence appeared unclear in our original submission. “Disagreement” should be more clearly defined as three or more panelists rating in each extreme. For example, if three panelists rated an indicator between 1-3 for appropriateness and the other seven panelists rated between 7-9, we would consider the indicator to have achieved a median score >7 but with disagreement. Therefore, the indicator would not advance to the next round due to the considerable dispersion in responses. Excluding items with disagreement (extreme ratings) ensures that we have conformity in our panel (i.e., only indicators with agreement among panelists will proceed). We have revised this statement to ensure comprehension.

Lines 131-133: As per RAM guidelines, indicators advanced to the second round if they achieved a median score between 7 and 9 for both criteria without disagreement, where disagreement was defined as three or more panelists rating in both extremes (i.e., 1-3 or 7-9) for either criterion [32].

4. Developing Technical Definition: what is the dealy between clinical/adm data is collected and processed, and assessible in ICES? How reevant are proposed datasets to determine quality indicators in real-life - limiting the delay in feedback as to make data relevant for QI efforts in the clinic?

Thanks for this inquiry. The ICES Data Dictionary lists all the datasets within the repository and specifies their date range, update frequency, and date of last update. ICES datasets corresponding to indicators endorsed through our process (e.g., the derived cohort of Congestive Heart Failure patients [CHF], Ontario Drug Benefit Claims [ODB], Ontario Health Insurance Plan [OHIP]) have been last updated in 2022 or 2023 and are updated quarterly, monthly, or annually. Therefore, there is only a minimal delay in which relevant clinical/administrative data is made available in ICES to examine these indicators. 

In an ongoing population-based retrospective study building upon findings from this consensus study, we are actively using the proposed datasets/technical definitions established through this work to examine family physician practice in Ontario, Canada. The technical definitions drafted, revised, and endorsed through this consensus process have shown to be highly relevant and feasible to measure indicators “in real life.” This work is alluded to in our Conclusion: 

Lines 393-398: We produced a measurable set of quality indicators that will support work to examine primary care provision for older adults using health administrative data. […] The technical definitions established through this study offer a case to trial their measurement within the context population-based data sources in Ontario, Canada.

In our Discussion (excerpt below), we describe how the intent of these endorsed indicators is not to monitor the performance of individual family physicians or offer “real time” feedback in clinic. As there are many providers, systems, and services that contribute to the primary health care of older adults, we do not intend to assess or report on the practice of individual clinicians. However, population-level measurement of these clinical activities may identify areas where high quality care for elderly patients is consistently provided and systematic challenges in delivering elderly-focused care. Measuring primary care quality for older adults using these metrics is a starting point to determine opportunities for resources, education, incentives, interventions, and policies to support quality improvement.

Lines 365-369: Quality measures do not function well on the individual (in this case, provider) level since other factors (e.g., setting, resources) may impact processes or resulting health outcomes. Therefore, we do not suggest that these indicators should be implemented in practice (such as to generate individual-level reports); rather, our findings may lead to education and awareness about approaches to measuring practice-based primary care encounters and for quality improvement.

5. Synchronous panel meeting: Did the second meeting, when experts reviewed and scored proposed quality indicators also done virtually in a panel format?

Thank you for seeking clarification. Each panelist virtually attended one synchronous panel meeting. Due to the limited availability of panelists, it was not possible to find a single meeting time that worked for everyone’s’ schedules – resulting in two separate panel meetings being held. We have since clarified this in the Results: 

Lines 276-277: Six panelists attended the first group meeting; four attended the second.

Following the group panel meetings, we summarized feedback, revised the technical definition workbook, and distributed the second questionnaire which was completed virtually/asynchronously over a two-week period. 

Results

6. Fig 1. Expand in text what the figure is supposed to represent - what is the conclusion?

Thanks for this suggestion. We have since summarized the literature review findings in-line. (Note: In response to your comment #11 below, the content of Fig 1 now appears in S1 Fig.) 

Lines 191-192: The search, screening, and extraction results of our literature review yielded 36 included texts and 500 candidate indicators (S1 Fig).

7. Fig 2, similar to above, expand the explanatory text in the manuscript.

We have improved our description of Fig 2 in the main text and provided a conclusion. (Note: In response to your comment #11 below, the content of Fig 2 now appears in S1 Fig.)

Lines 196-197: We organized the 282 quality indicators by the 18 FM-COE Priority Topics and streamlined indicators within each Topic (S1 Fig), resulting in 178 items for internal screening.

8. Technical panel, line 190 and ff. Move descriptive part to Methods, where Provider characteristics are presented.

Thanks for this feedback. We considered your suggestion but have decided to keep the descriptive characteristics of our panelists under this “Technical expert panel” heading in our Results. This paragraph is similar to presenting the descriptive characteristics of study participants that often appears at the beginning of a Results section for an observational, experimental, or qualitative study. Whereas we describe the eligibility criteria of prospective panelists in our Methods, we present the characteristics of those we recruited and engaged in our Results. 

9. Consider reducing number of tables/figures.

Thank you for this suggestion. In response to your comment #11 below, where we describe how we reduced the number of tables, figures, and supplementary materials. 

Discussion

10. Statement that consensus was established - line 292. Provide data support for the statement. Moreover, discuss relatively low median scores for both Appropriateness and Importance (5-7). Sounds like these quality indicators are not rated very high by neither providers, nor technical experts in terms of Appropriateness and Importance.

We appreciate this suggestion. Our introductory statement in the Discussion stating that consensus was established is supported by our Results presented under “Round #2 questionnaire” (excerpt to follow). Further, a supplementary material (S5 Table) lists the median scores of all 15 indicators on both criteria from the second questionnaire – resulting in the endorsement of 12 indicators. We summarize these raw scores resulting in the establishment of our final indicator set in the main text: 

Lines 287-289: Panelists rated 15 quality statements and their corresponding technical definitions in the second questionnaire. Median scores ranged from 6.0 to 9.0 for appropriateness and 6.5 to 8.5 for importance (S5 Table). 12 indicators met our threshold for inclusion in the final indicator set (Table 2); three indicators were eliminated.

In our first questionnaire, median scores ranged from 5.5 to 9.0 for appropriateness and 6.0 to 9.0 for importance. In the second questionnaire, median scores ranged from 6.0 to 9.0 for appropriateness and 6.5 to 8.5 for importance. As you noted, some indicators received relatively lower median scores and were subsequently excluded from future rounds (as per our thresholds:

• Questionnaire 1: 6 indicators did not have a median score between 7-9 for both criteria without disagreement and were excluded. 

• Questionnaire 2: 3 indicators were rated less than 7-9 on both criteria by more than 60% of panelists and were excluded. 

In response to your feedback, we have since added the following to our Discussion: 

Lines 317-319: The endorsed statements express four of 18 FM-COE Priority Topics and achieved increasingly greater ratings on “appropriateness” and “importance” in subsequent rounds.

11. The manuscript contains a substantial amount of data, specifically in a rather large number of attachments. This data is only limited referred to in the result section. moreover, even less discussed in the discussion section. Rather, generic comments are provided, e. g., line 307 and 308: "-were concentrated on biomedical aspects of caring for elderly patients - namely, the "Medical Conditions". This is well known, lack of functional quality indicators, survival data, and patient-perspectives are major limitations in current quality of care indicator data. as in this article, much data is process rather than outcome focused.

We appreciate your comments on this and have since reduced the number of tables, figures, and supplementary materials by five; we eliminated three figures and two supplementary materials. 

We kept two tables in the main text as they convey pertinent information: Table 1 describes the criteria used to screen candidate indicators, while Table 2 lists our main findings (the endorsed quality statements).

We combined Fig 1, Fig 2, and Fig 3 into a single figure, and moved them as a supplementary material (now S1 Fig). While the literature search and screening process was important to establish our questionnaire, the level of detail in these figures is not integral to our main results. 

The S1 Table (reporting guideline checklist) and S2 Table (summary of included literature) remain. 

We removed S3 Table it was not necessary to illustrate the process of how candidate indicators were grouped thematically. We also removed S4 Table as the quality statements assessed in the round 1 questionnaire appear in S5 Table (where the ratings for each are presented). Therefore, the original S5 Table (now S3 Table) remains which presents the ratings from the first questionnaire. We kept an example of a quality statement (originally S6 Table; now S4 Table) so readers understand the level of detail in our technical definitions. S5 Table is necessary as it presents ratings from round 2. Lastly, S6 Table lists items that we did not present in the panel group meeting and were not rated in the second questionnaire, but can be re-examined in future work. 

 **

Thanks for your feedback about our level of detail in the Results and Discussion sections. 

In our Results, we present the ratings on both criteria for indicators and summarize open-ended feedback from panelists to elucidate their ratings in both questionnaires. We describe the specific datasets that were used to establish our technical definitions for endorsed indicators. Our flow diagram of endorsed indicators organized by the FM-COE Priority Topics is a useful graphic illustrating the inclusion/exclusion of indicators at each stage. We believe the level of detail in this section is sufficient and complements information presented in our tables, figures, and supplements.

However, as per your feedback, we have since presented more specific results in our Discussion. 

Lines 317-319: The endorsed statements express four of 18 FM-COE Priority Topics and achieved increasingly greater ratings on “appropriateness” (round 1: 5.5 to 9.0; round 2: 6.0 to 9.0) and “importance” (round 1: 6.0 to 9.0; round 2: 6.5 to 8.5) in subsequent rounds.

Lines 331-333: This process revealed that many indicators – those identified from the literature and endorsed through our consensus process – were concentrated on the biomedical aspects of caring for elderly patients – namely, the “Medical Conditions” (n=4) and “Appropriate Prescribing” (n=5) FM-COE Priority Topics.

**

Lastly, you commented, “much data is process rather than outcome focused.” In our Introduction, we indicate that this study focuses on process metrics (rather than structures or outcomes) as per the Donabedian model. We rationalize this in our published study protocol (excerpt below). 

Lines 76-78: We focus on quality indicators that are directly relevant to the clinical activities of physicians (practice-based) and classified as acts of healthcare service delivery (processes) as per the Donabedian model [21,24].

Excerpt from published study protocol (Correia, R. H., Siu, H. Y. H., Vanstone, M., Jones, A., Gopaul, A., & Costa, A. P. (2023). Development of practice-based quality indicators for the primary care of older adults: a RAND/UCLA Appropriateness Method study protocol. BMJ Open, 13(9), e072232): 

The Donabedian model—the dominant quality improvement paradigm in health services research—conceptualises and enables evaluations of the quality and performance of medical care through three inter-related components: structures, processes and outcomes.20 34 35 Structures are static or technical factors affecting the context in which medical care is delivered (eg, human resources, payment models and healthcare institutions), processes refer to acts of healthcare delivery (eg, diagnoses, treatments, preventative care and patient education) and outcomes include the effects of medical care on patient health (eg, prognosis, patient satisfaction and health service utilisation). In this study, we focus on indicators that can be classified as processes, given their relevance to the practice-based clinical activities of physicians. Structures will not be examined due to their upstream and evolving nature, which is challenging to discern from individual-level data. Outcomes are influenced by multiple structures and processes, including care delivered by multidisciplinary providers in different settings, which is not specific to the care of family physicians.

12. The Discussion section is very limited, does not refer to the greater debate of relevance, and hardly et al refer to other studies of relevance. Furthermore, the authors should discuss the relevance of using only expert-derived quality indicators in an age of patient-centered primary care. 

Thank you for this feedback. We have since strengthened our Discussion by adding two additional paragraphs. First, we unpack the limitation of not including patients in this study but discuss how their involvement in future work will further our understanding of quality primary care for older adults (which also address your comment #13 below). Second, we discuss the challenges of relying on administrative data to measure quality and the generalizability of our indicator set. 

Lines 341-361: Further, our panelists with demonstrated clinical and research expertise likely prioritized indicators differently than older adults or informal caregivers would have. Patients are increasingly involved in quality improvement initiatives and their first-hand experiences and inputs into problem-solving have demonstrated value in redesigning care aligned with patient priorities [52–54]. For example, indicators relevant to “organizing care using community resources,” a FM-COE Priority Topic, were identified in our literature review but not endorsed. Therefore, by not including patients or caregivers on our expert panel, the endorsed indicators may only constitute some of the primary care activities that are important or meaningful to patients. Through a separate but complementary study, we will engage older adults to compare and establish patient-important indicators. This approach will extend how we conceptualize and assess quality primary care for older adults, and maximize patient perspectives and feedback in an open-ended format, rather than limiting their viewpoints to the constraints of feasible secondary data.

Measurable quality indicators are limited by information sources available and technical definitions – which are necessary to operationalize indicators – pertain to particular settings, thereby, affecting their generalizability [21]. There is substantial diversity in how clinical scenarios and activities are defined in different healthcare contexts and reported in administrative data sources, which prohibits the development of universal technical definitions. In an effort to establish measurable indicators to support future work, technical definitions developed through this process are contextualized to community-based primary care settings in Ontario, Canada, and relevant information sources. While the specific data set and variable names of our endorsed indicators may not be directly apply to other information sources, they can be adapted to the local context if studied in other settings [55].

13. Lack of including patients or caregivers are briefly touched on in Limitation (ll 340-343. However, referring to another study to be, does not address this major shortcomings. Even when exploring how to use administrative data. such data can be generated for patients and caretakers as well.

Thank you for highlighting this important limitation. The following excerpt from our published study protocol (particularly, the bolded sentences) may help explain why a separate study to engage patients and caregivers was warranted. 

Excerpt from published study protocol (Correia, R. H., Siu, H. Y. H., Vanstone, M., Jones, A., Gopaul, A., & Costa, A. P. (2023). Development of practice-based quality indicators for the primary care of older adults: a RAND/UCLA Appropriateness Method study protocol. BMJ Open, 13(9), e072232): 

…[T]hrough a separate study supported by the Transdisciplinary Understanding and Training on Research-Primary Healthcare (TUTOR-PHC) Patient and Community Engagement Research Fellowship, we intend to engage older adults to understand factors impacting the quality of primary care provision. This complementary study aims to compare aspects of primary care practice that are important to patients with indicators derived from the RAM study. We chose to engage the public through a separate consultative approach to eliminate technical discussions about operationalising indicators using health administrative data. This approach maximises patient perspectives and feedback in an openended format, rather than limiting their viewpoints to the constraints of feasible secondary data. This independent activity will allow the public to rate indicators identified in our formal consensus process and provide perspectives on what was created.

Since the focus of this study was to ultimately establish a measurable set of quality indicators, panelists reviewed technical concepts (e.g., diagnostic codes, physician fee codes, drug identification numbers) in our group panel meeting and rated these in our questionnaires. We devised eligibility criteria for clinicians and researchers constituting our technical expert panel to ensure they possess an in-depth understanding of clinical activities and health services research. As the validity of our consensus study findings depended on the expertise of panellists, we determined it was important to recruit clinicians and health services researchers with specific qualifications (e.g., clinical years in practice, research publications) to demonstrate their expertise on the candidate indicators.

We recognize the value of patient and public engagement in health services research, and believe patients’ first-hand experiences of seeking primary care will benefit our conceptualization of what high-quality primary care constitutes for older adults. However, we chose to engage patients and caregivers through a separate collaborative process where their ideas will not be constrained by the RAM questionnaire items and reliance on technical/measurable aspects of care. For example, through this study, we identified that many endorsed indicators reflect the biomedical FM-COE Priority Topics (e.g., medical conditions, appropriate prescribing) that are more readily available in administrative data. In contrast, we hypothesize that the FM-COE Priority Topics that are more challenging to measure using administrative data will be highly regarded by patients and caregivers (e.g., communication, family and informal care supports, organizing care using community resources). We look forward to examining this hypothesis in our separate study where we will compare indicators of high-quality primary care that were endorsed by experts through this process with those that patients value. Further, this separate study will describe older patients’ needs and preferences for high-quality primary care and, ultimately, highlight aspects of family physician practice that patients value most. We are actively working with patient partners to co-design this study. In future work, we will also examine opportunities to measure patient-important indicators within administrative data. 

As detailed above in response to comment #12, we have added a paragraph to our Discussion about the value of patient engagement in this work. 

Lines 341-352: Further, our panelists with demonstrated clinical and research expertise likely prioritized indicators differently than older adults or informal caregivers would have. Patients are increasingly involved in quality improvement initiatives and their first-hand experiences and inputs into problem-solving have demonstrated value in redesigning care aligned with patient priorities [52–54]. For example, indicators relevant to “organizing care using community resources,” a FM-COE Priority Topic, were identified in our literature review but not endorsed. Therefore, by not including patients or caregivers on our expert panel, the endorsed indicators may only constitute some of the primary care activities that are important or meaningful to patients. Through a separate but complementary study, we will engage older adults to compare and establish patient-important indicators. This approach will extend how we conceptualize and assess quality primary care for older adults, and maximize patient perspectives and feedback in an open-ended format, rather than limiting their viewpoints to the constraints of feasible secondary data.

Reviewer #3:

Thank you for the opportunity to review this manuscript. The study is well conducted and provides an interesting view on how to characterize quality of care for older primary care patients on population level. Using RAND/UCLA appropriateness study practice-based quality measures are derived to measure quality of care for older adults in primary care on population level. Overall, this is a well-written and interesting paper. The protocol paper has been published earlier.

Some questions and minor comments could enhance the readability of the paper.

Introduction

1. Please clarify what you mean with ‘practice-based process metrics’

Thanks for this comment. We acknowledge that we did not provide a definition of ‘practice-based process metrics’ in our original submission; a definition has been subsequently added. 

Lines 76-78: We focus on quality indicators that are directly relevant to the clinical activities of physicians (practice-based) and classified as acts of healthcare service delivery (processes) as per the Donabedian model [21,24].

2. An operationalization of quality of care Is not described, please elaborate how this concept is viewed by the authors.

Thanks for this comment. In our published study protocol, we describe how we operationalized quality of care in this study. 

Excerpt from published study protocol (Correia, R. H., Siu, H. Y. H., Vanstone, M., Jones, A., Gopaul, A., & Costa, A. P. (2023). Development of practice-based quality indicators for the primary care of older adults: a RAND/UCLA Appropriateness Method study protocol. BMJ Open, 13(9), e072232): 

The Donabedian model—the dominant quality improvement paradigm in health services research—conceptualises and enables evaluations of the quality and performance of medical care through three inter-related components: structures, processes and outcomes.20 34 35 Structures are static or technical factors affecting the context in which medical care is delivered (eg, human resources, payment models and healthcare institutions), processes refer to acts of healthcare delivery (eg, diagnoses, treatments, preventative care and patient education) and outcomes include the effects of medical care on patient health (eg, prognosis, patient satisfaction and health service utilisation). In this study, we focus on indicators that can be classified as processes, given their relevance to the practice-based clinical activities of physicians. Structures will not be examined due to their upstream and evolving nature, which is challenging to discern from individual-level data. Outcomes are influenced by multiple structures and processes, including care delivered by multidisciplinary providers in different settings, which is not specific to the care of family physicians.

The additional sentence added to our Introduction in response to your previous comment (#1 above) clarifies that we operationalize quality of care in this work by focusing on processes as per the Donabedian model: 

Lines 76-78: We focus on quality indicators that are directly relevant to the clinical activities of physicians (practice-based) and classified as acts of healthcare service delivery (processes) as per the Donabedian model.[21,24]

Methods

3. The steps are easy to follow. Hence, please clarify the following aspects: Table 1 point 4 “practice-based orientation.

Thank you for asking that we clarify what is meant by practice-based orientation. In response to your previous comment (#2 above), we have since defined what is meant by “practice-based.” We revised the text in Table 1 point 4 to further clarify what is meant: 

Table 1: Is the indicator practice-based? Is the indicator directly relevant to clinical activities/practice?

Results

4. Round 1 questionnaire please describe the six indicators that were eliminated

Thanks for this suggestion. We have since described the six indicators that were eliminated. 

Lines 224-226: Indicators that did not advance were rated poorly for not pertaining to the role/scope of family physicians or describing inappropriate clinical activities or timeframes.

5. Many quotes from the panelists are described, what was done with this information?

Thank you for asking what was done with the qualitative data collected as open-ended feedback in our questionnaires. While we summarize panellists’ open-ended responses in the Results section, we acknowledge that we did not clearly describe what was done with the qualitative data in our Methods. We have since added this: 

Line 181: Qualitative data were used to revise quality statements and technical definitions.

As detailed in the Results, we describe how panelists suggested revisions to the wording of quality statements and the proposed technical definitions, such as by referencing updated clinical practice guidelines or medication lists. The following excerpt illustrates this: 

Lines 309-313: Panelists also referenced guidelines, reports, research articles, and other documents to refine specifications, and suggested revisions to how some quality statements were stated. For example, 

“To be future oriented, should the item read THEN "the PCP should perform tests aligned with the MOST CURRENT Canadian Consensus on Dementia". Presumably, the science/evidence will evolve and this language will prevent the indicator from becoming stale dated.” – Panelist #5, Indicator #9 

6. How many panelists were participating during each session?

We agree it is important to specify how many panelists attended either of the two group meetings. We have since clarified this in our Results. 

Lines 276-277: Six panelists attended the first group meeting; four attended the second.

Discussion

7. Please elaborate more why and how the selection was done to reduce the number of indicators – only time limitation is somewhat vague.

Thanks for these suggestions. When we originally set our threshold for indicators to advance from the first to second round, we were not expecting so many indicators to be rated so highly. (In retrospect, we should have raised our rating threshold to exclude more indicators from passing beyond the first round.) This resulted in having far too many indicators (n=55) to feasibly present/discuss in our previously scheduled panel group meetings, which were allotted for 2 hours in our original letter of information to panelists. 

The four excerpts below from our Methods, Results, and Discussion address why and how indicators were prioritized for discussion/rating. 

[Methods – “how” we prioritized indicators advancing from round 1 to 2]

Lines 161-163: Indicators were presented for discussion according to rank order; limited time and resources necessitated prioritizing a subset of indicators to present in each meeting.

[Results– “how” and “why” we prioritized indicators advancing from round 1 to 2]

Lines 249-258: From the 55 endorsed quality statements, it was necessary to develop technical definitions, an example of which is provided in S6 Table. We rank-ordered the endorsed indicators from the first questionnaire to pragmatically determine a subset that would be reasonable to present in our allotted panel meeting times; it would not have been possible to discuss all 55 endorsed indicators during the predetermined two-hour timeframe. We considered what threshold of the most highly rated indicators would be feasible to present (e.g., top 10% = 6 indicators, top 25% = 14 indicators, etc.). We allotted 10 minutes to review the project objectives and facilitate introductions, 15 minutes to review ratings from the first questionnaire, and five minutes per indicator to gather feedback on the proposed technical definition – resulting in sufficient time to present 19 indicators (the top 35% most highly rated indicators) over 95 minutes. 

Our approach of rank-ordering indicators (based on those rated most highly) enabled us to feasibly present as many endorsed indicators as possible in our allotted panel group meetings. We also acknowledge this in our Limitations and suggest an opportunity for future work to re-engage panelists to seek feedback on the endorsed indicators that were not presented in the panel group meeting: 

[Discussion – “how” and “why” we prioritized indicators advancing from round 1 to 2]

Lines 383-388: We only presented a subset of endorsed indicators from the first questionnaire and their technical definitions in the group panel meetings. Although excluding indicators for pragmatic reasons may have biased our findings, our study demonstrated the utility of using a consensus process to derive a measurable indicator set for those rated most highly. We did not initially set out to establish a comprehensive indicator set; rather, we aimed to determine whether it was possible to establish consensus on measurable indicators by referencing administrative data holdings [21].

Lines 390-391: If a useful profile can be created, we can re-engage panelists to solicit feedback on the outstanding indicators (S6 Table).

8. Please elaborate on the generalizability of the available administrative data

Thank you for this feedback. We acknowledge that we discussed the generalizability of the administrative data at length in our published study protocol, but did not detail this sufficiently in our original submission of this manuscript. 

We believe the novelty of our work is in establishing measurable quality indicators using the case of population-based data holdings at ICES in Ontario. In contrast to much of the quality improvement literature that does not provide measurable indicators, we aimed to produce highly feasible technical definitions to then examine the endorsed indicators; albeit, at the expense of establishing generalizable indicators that pertain across settings. While we recognize the limitations of establishing measurable indicators within a particular setting/context, our work examines the feasibility of establishing measurable definitions within this case that can then be adapted elsewhere. 

There is much diversity (provincially, nationally, internationally) in how health administrative data are collected and made available to researchers, which means the specific technical definitions we developed (e.g., dataset names, variable names, timings) will only apply within the ICES context. In order to be measurable, we had to select a case (i.e., ICES) to reference administrative data holdings to draft our technical definitions. However, there is a high probability that the operational definitions we develop can be adapted to other contexts for meaningful research and evaluation. 

We have since elaborated on these strengths/limitations regarding generalizability in our Discussion. 

Lines 353-361: Measurable quality indicators are limited by information sources available and technical definitions – which are necessary to operationalize indicators – pertain to particular settings, thereby, affecting their generalizability [21]. There is substantial diversity in how clinical scenarios and activities are defined in different healthcare contexts and reported in administrative data sources, which prohibits the development of universal technical definitions. In an effort to establish measurable indicators to support future work, technical definitions developed through this process are contextualized to community-based primary care settings in Ontario, Canada, and relevant information sources. While the specific data set and variable names of our endorsed indicators may not be directly apply to other information sources, they can be adapted to the local context if studied in other settings [55].

---

## [Decision Letter · Decision Letter 1]

25 Dec 2023

PONE-D-23-32654R1Primary care quality for older adults: practice-based quality measures derived from a RAND/UCLA appropriateness method studyPLOS ONE

Dear Dr. Correia,

Thank you for submitting your manuscript to PLOS ONE. After careful consideration, we feel that it has merit but does not fully meet PLOS ONE’s publication criteria as it currently stands. Therefore, we invite you to submit a revised version of the manuscript that addresses the points raised during the review process.

We look forward to receiving your revised manuscript.

Kind regards,

Mohammad Sadegh Taghizadeh

Academic Editor

PLOS ONE

Reviewers' comments:

Reviewer's Responses to Questions

**Comments to the Author**

1. If the authors have adequately addressed your comments raised in a previous round of review and you feel that this manuscript is now acceptable for publication, you may indicate that here to bypass the “Comments to the Author” section, enter your conflict of interest statement in the “Confidential to Editor” section, and submit your "Accept" recommendation.

Reviewer #1: All comments have been addressed

Reviewer #2: All comments have been addressed

2. Is the manuscript technically sound, and do the data support the conclusions?

Reviewer #1: Yes

Reviewer #2: (No Response)

3. Has the statistical analysis been performed appropriately and rigorously? 

Reviewer #1: I Don't Know

Reviewer #2: (No Response)

4. Have the authors made all data underlying the findings in their manuscript fully available?

Reviewer #1: Yes

Reviewer #2: (No Response)

5. Is the manuscript presented in an intelligible fashion and written in standard English?

Reviewer #1: Yes

Reviewer #2: (No Response)

6. Review Comments to the Author

**Reviewer #1: **

This revision is improved, notably from a more robust discussion of limitations and methodological focus. The authors responded well to all reviewer comments.

There are remaining points/questions as below.

-Line 28: I wouldn't say pan-Canadian. 9 were based in Ontario. Perhaps just Canadian

-Line 78: Donabedian model should be defined here (in the author's own words)

-Line 259: to do what?

-Table 2:

--why not have 1 and 4 (vaccines) next to each other?

--2 and 12 can both go under cognitive impairment, yes?

--why are rows from Medical Conditions shaded and most rows from Appropriate prescribing shaded (and the others not)?

-Line 353: comma after available

-Line 384: "pragmatic reasons" is the predetermined two-hour panel meeting length, yes? If so, I would specify that to acknowledge it as a limitation "pragmatic reasons (i.e., the predetermined two-hour panel meeting length)"

-Figures (Figure 1, S1-2, and scatter plot figures as below) should have figure legends.

Supplementary Figure 2:

--x-axis: round instead of group, yes?

--perhaps a more meaningful title than response_bias_score? Isn't this the "Wilcoxon matched-pairs signed-rank test for consensus scores"?

--including n number for each round would be nice and clear

--how did the Pr > Z scores translate to the p-value? wouldn't these be the same?

-Supplementary Table 6: This title needs changed, correct? Items in the "Presented in panel group meeting" and "Rated in questionnaire 2" columns were prioritized in round 2, correct? The table should also be referenced accordingly in the results section, not just at the end of the discussion

-While I acknowledge Reviewer #2's comment to reduce the number of tables/figures, I believe they are all meaningful in this revision and describe important study components. I feel the additional scatter plot figure in response to Reviewer #1 provides meaningful context to Figure 1 and Supplementary Table 6 and should be included in this submission (and would also be more "outcome focused" aligning with Reviewer #2's comments). With this though, I don't see where it was mentioned in the manuscript where 14 indicators had equal summed scores and had to be sequenced by lowest IQR. A similar scatter plot with IQR on the Y axis should thus also be included (perhaps part of the same figure as the summed median score scatter plot) for these 14 items to clearly see the raw IQR difference between the 5 included vs. other items. Were they sequenced based on summed IQR (Appropriateness + Importance)? This should be included as a result and limitation.

-I do agree with Reviewer #3 that "operationalize" is not explicitly described. I interpret it to mean able to be examined data-wise, but I could also (I believe mis-)interpret it to mean directly used to clinical care. Can "operationalize" be described in the introduction? perhaps with the Donabedian model? or changed to "examine" if appropriate?

**Reviewer #2: **

The authors have addressed my concerns. In the few cases they have not, they have provided a good rationale for not doing so.

7. PLOS authors have the option to publish the peer review history of their article (what does this mean?). If published, this will include your full peer review and any attached files.

Reviewer #1: No

Reviewer #2: **Yes: **Bengt B. Arnetz

---

## [Author Response · Author response to Decision Letter 1]

3 Jan 2024

Dear PLOS ONE Editorial Office and Reviewers:

We appreciate your continued support in strengthening our manuscript. Thank you for providing encouraging and constructive feedback. In our Response to Reviewers, we describe the changes made to our manuscript to address the outstanding peer review comments. All suggestions and comments from the reviewer have been incorporated. 

Thank you again for your consideration.

Sincerely, 

Rebecca Correia

***

1. Line 28: I wouldn't say pan-Canadian. 9 were based in Ontario. Perhaps just Canadian

Thanks for this comment; we have revised accordingly.

Line 27-29: We conducted a two-round RAND/UCLA Appropriateness Method (RAM) study and recruited 10 Canadian clinicians and researchers with expertise relevant to the primary care of elderly patients.

2. Line 78: Donabedian model should be defined here (in the author's own words)

Thanks for this suggestion. We have since added the bolded sentence to define the Donabedian model in the following excerpt: 

Lines 78-82: We focus on quality indicators that are directly relevant to the clinical activities of physicians (practice-based) and classified as acts of healthcare service delivery (processes). Processes are one of three interrelated components comprising the Donabedian model – the dominant quality improvement paradigm that enables evaluations of medical care quality and the performance of health systems [21,24].

3. Line 259: to do what?

Thanks for this comment. Based on your comment, we have re-sequenced content under the “Technical definitions” sub-heading to clarify “what was done” with the indicators/technical definitions during the panel meeting. This excerpt now better reflects how we reviewed and gathered feedback on the technical definitions in our panel group meetings: 

Lines 256-327:

Technical definitions

From the 55 endorsed quality statements, it was necessary to develop technical definitions, an example of which is provided in S4 Table. We documented our efforts to operationalize each quality statement and posed discussion questions within the technical definition workbook used to guide our discussion in the synchronous panel meeting. For example, if the frequency of a health service encounter or timeframe for follow-up was not specified within the quality statement, we noted these outstanding aspects as points of discussion. 

We referenced four ICES datasets […] 

We rank-ordered the endorsed indicators from the first questionnaire to pragmatically determine a subset that would be reasonable to present in our allotted panel meeting times; it would not have been possible to discuss all 55 endorsed indicators during the predetermined two-hour timeframe. We considered what threshold of the most highly rated indicators would be feasible to present (e.g., top 10% = 6 indicators, top 25% = 14 indicators, etc.). We allotted 10 minutes to review the project objectives and facilitate introductions, 15 minutes to review ratings from the first questionnaire, and five minutes per indicator to gather feedback on the proposed technical definition – resulting in sufficient time to review 19 indicators (the top 35% most highly rated indicators) over 95 minutes (S5 Table; S2 Fig). […]

Synchronous panel meeting

We gathered feedback on the technical definitions during our synchronous panel meetings. […]

Through these discussions, we unanimously omitted four quality statements; two items were duplicates and two were not possible to accurately specify using administrative data. Panelists also suggested revising the language of eleven quality statements; these revised statements were incorporated into the second questionnaire and rated. Following the meetings, we finalized our technical definition workbook to reflect group feedback. 

4. Table 2:

--why not have 1 and 4 (vaccines) next to each other?

--2 and 12 can both go under cognitive impairment, yes?

--why are rows from Medical Conditions shaded and most rows from Appropriate prescribing shaded (and the others not)?

We appreciate your suggestions to improve Table 2 (Line 338). We moved indicators #1 and #4 (those pertaining to vaccines) next to each other, as you suggested. Indicators #2 (now #4) and #12 are also relevant to Cognitive Impairment, as you said; we have since added a footnote to denote this. Lastly, we regret that the shading of rows in this table was incorrect in our previous submission. No rows should have been shaded/filled; we since have corrected this. 

5. Line 353: comma after available

Thanks for catching this grammatical error. We have since added a comma.

Line 412: Measurable quality indicators are limited by information sources available, and technical definitions – which are necessary to operationalize indicators – pertain to particular settings, thereby, affecting their generalizability [21].

6. Line 384: "pragmatic reasons" is the predetermined two-hour panel meeting length, yes? If so, I would specify that to acknowledge it as a limitation "pragmatic reasons (i.e., the predetermined two-hour panel meeting length)"

Thanks for this suggestion; we have revised accordingly. 

Lines 443-446: Although excluding indicators for pragmatic reasons (i.e., the predetermined two-hour panel meeting length) may have biased our findings, our study demonstrated the utility of using a consensus process to derive a measurable indicator set for those rated most highly.

7. Figures (Figure 1, S1-2, and scatter plot figures as below) should have figure legends.

Thanks for these suggestions. We have added figure legends to Fig 1, S1 Fig, S2 Fig, and S3 Fig:

Lines 354-357: Fig 1. Flow diagram of endorsed indicators, by FM-COE Priority Topic

Legend: 61 indicators across 10 FM-COE Priority Topics were identified in the literature review for rating; 55 were endorsed in the first questionnaire. Of the 19 indicators reviewed in the panel meeting, 12 indicators across four Priority Topics were endorsed in the second questionnaire. 

Lines 662-673: S1 Fig. Literature search, organization, and screening results

Legend: Steps 1 to 4 outline the literature search results of 282 candidate indicators. Steps 5 to 7 illustrate how the candidate indicators were organized by the FM-COE Priority Topics and streamlined into 178 quality statements. Steps 8 and 9 display the results of internal feasibility screening and clinical review, resulting in 61 quality statements for inclusion in the first questionnaire.

S2 Fig. Distribution of summed median scores and IQRs for indicators that advanced from Round 1

Legend: 19 indicators whose summed median scores were rated most highly were prioritized for discussion in the group panel meeting. 14 indicators had equal summed median scores and were rank-ordered by the lowest summed IQR for prioritization. 

S3 Fig. Distribution of Wilcoxon scores (matched-pairs signed-rank test)

Legend: The p-value obtained by the Wilcoxon matched-pairs signed-rank test suggests a significant difference in consensus scores between the two RAM rounds.

8. Supplementary Figure 2:

--x-axis: round instead of group, yes?

--perhaps a more meaningful title than response_bias_score? Isn't this the "Wilcoxon matched-pairs signed-rank test for consensus scores"?

--including n number for each round would be nice and clear

--how did the Pr > Z scores translate to the p-value? wouldn't these be the same?

Thanks for these suggestions. We have incorporated the first three suggestions you provided above into S3 Fig.

In response to your fourth comment, the Wilcoxon matched-pairs signed-rank test produces two estimates (Pr > Z and Pr > |Z|) which relate to p-values. We chose to report and interpret the value 0.1303 (Pr > |Z|) in our main text, which represents the p-value associated with the absolute value of the Z statistic. We selected Pr > |Z| as it considers extreme values in both directions; we were not only interested in deviations in a specific direction. However, both Pr > Z and Pr > |Z| appear in S3 Fig and can be interpreted as exceeding the significance threshold of 0.05, suggesting the null hypothesis is not rejected.

9. Supplementary Table 6: This title needs changed, correct? Items in the "Presented in panel group meeting" and "Rated in questionnaire 2" columns were prioritized in round 2, correct? The table should also be referenced accordingly in the results section, not just at the end of the discussion

Thanks for this suggestion. We have revised the title of S6 Table to be, “Prioritization of endorsed items from round 1 to round 2,” and have also referenced it in our Results on Line 311. Due to referencing this table earlier in our main text, please note that Table S6 in our previous submission is now Table S5. 

10. While I acknowledge Reviewer #2's comment to reduce the number of tables/figures, I believe they are all meaningful in this revision and describe important study components. I feel the additional scatter plot figure in response to Reviewer #1 provides meaningful context to Figure 1 and Supplementary Table 6 and should be included in this submission (and would also be more "outcome focused" aligning with Reviewer #2's comments). With this though, I don't see where it was mentioned in the manuscript where 14 indicators had equal summed scores and had to be sequenced by lowest IQR. A similar scatter plot with IQR on the Y axis should thus also be included (perhaps part of the same figure as the summed median score scatter plot) for these 14 items to clearly see the raw IQR difference between the 5 included vs. other items. Were they sequenced based on summed IQR (Appropriateness + Importance)? This should be included as a result and limitation.

Thanks for affirming the importance of the additional scatter plot added in response to Reviewer #1’s previous feedback. We have since added this scatter plot as S2 Fig (Line 311). We have added a plot with summed IQRs on the y-axis to display the differences in summed IQRs for the 14 indicators with equal summed scores. 

In the following excerpt from our Methods, we describe how we sequenced indicators with equal summed scores by the lowest summed IQR for appropriateness and importance: 

Lines 141-146: Among indicators deemed eligible for inclusion in the second round, we rank-ordered the indicators to develop corresponding technical definitions efficiently and feasibly. We summed the median scores for each indicator on both criteria and computed interquartile ranges (IQRs). We sequenced indicators by those with the highest summed median score. For indicators with equal summed median scores, we prioritized those with the lowest summed IQR.

In addition to including these scatterplots in S2 Fig and referring to them in our Methods, we have also added the following to our Results and Limitations: 

Lines 314-316: In round 1, 14 indicators were rated most highly with summed median scores ranging from 18 to 16.5. 14 indicators achieved the next highest summed median score of 16; from these, 5 were prioritized based on having the lowest summed IQR.

Lines 446-448: We prioritized indicators for discussion in the panel meeting by rank-ordering endorsed indicators from round 1 with the highest summed median scores and, for those tied, the lowest summed IQR.

11. I do agree with Reviewer #3 that "operationalize" is not explicitly described. I interpret it to mean able to be examined data-wise, but I could also (I believe mis-)interpret it to mean directly used to clinical care. Can "operationalize" be described in the introduction? perhaps with the Donabedian model? or changed to "examine" if appropriate?

Thanks for this suggestion. We have since defined “operationalize” in our Introduction, as per the bolded excerpt below.

Lines 71-76: Research focused on the development of elderly-focused quality indicators for primary care to-date have not been concerned with operationalizing them [21]. In this context, “operationalize” refers to developing technical specifications to apply, measure, and examine indicators using available data holdings. Despite the limitations of using secondary data to make inferences about quality [21], indicators developed without reference to information sources or those that are not feasible to operationalize within a particular context are limited in their utility and capacity to promote change.

---

## [Decision Letter · Decision Letter 2]

8 Jan 2024

Primary care quality for older adults: practice-based quality measures derived from a RAND/UCLA appropriateness method study

PONE-D-23-32654R2

Dear Dr. Correia,

We’re pleased to inform you that your manuscript has been judged scientifically suitable for publication and will be formally accepted for publication once it meets all outstanding technical requirements. **Specifically, please apply the reviewer's comments in the proof version.**

Kind regards,

Mohammad Sadegh Taghizadeh

Academic Editor

PLOS ONE

Additional Editor Comments (optional):

Reviewers' comments:

Reviewer's Responses to Questions

**Comments to the Author**

1. If the authors have adequately addressed your comments raised in a previous round of review and you feel that this manuscript is now acceptable for publication, you may indicate that here to bypass the “Comments to the Author” section, enter your conflict of interest statement in the “Confidential to Editor” section, and submit your "Accept" recommendation.

Reviewer #1: All comments have been addressed

2. Is the manuscript technically sound, and do the data support the conclusions?

Reviewer #1: Yes

3. Has the statistical analysis been performed appropriately and rigorously? 

Reviewer #1: I Don't Know

4. Have the authors made all data underlying the findings in their manuscript fully available?

Reviewer #1: Yes

5. Is the manuscript presented in an intelligible fashion and written in standard English?

Reviewer #1: Yes

6. Review Comments to the Author

Reviewer #1: The authors have responded to all reviewer suggestions in a very thoughtful, generally thorough, and timely fashion.

A few small editorial points as below, but I believe this article is now scientifically fine to be accepted pending statistical review (specifically confirming that the Figure S3 Pr > |Z| value of 0.1303 aligns with the p value of 0.0391 reported on line 304 of this submission)..

--lines 80-81: The Donabedian model is acceptably characterized but not optimally defined here. Perhaps "Along with clinical structures and medical outcomes, healthcare service delivery processes are one of three interrelated components..."

--lines 306-309: Figure 1 legend will need reformatted as it is currently included in the text rather than at the end of the manuscript with the other figure legends..

--lines 374-375: is this meant to read "quality primary care quality"?

--line 411: is this meant to read "the context population-based" or "the context of population-based"?

7. PLOS authors have the option to publish the peer review history of their article (what does this mean?). If published, this will include your full peer review and any attached files.

Reviewer #1: No

---

## [Editor Report · Acceptance letter]

11 Jan 2024

PONE-D-23-32654R2 

PLOS ONE

Dear Dr. Correia, 

I'm pleased to inform you that your manuscript has been deemed suitable for publication in PLOS ONE. Congratulations! Your manuscript is now being handed over to our production team.

Kind regards, 

on behalf of

Dr. Mohammad Sadegh Taghizadeh 

Academic Editor

PLOS ONE